# An aspartyl protease defines a novel pathway for export of *Toxoplasma* proteins into the host cell

Michael J Coffey[1,2], Brad E Sleebs[1,2], Alessandro D Uboldi[1,2], Alexandra Garnham[1,2], Magdalena Franco[3], Nicole D Marino[3], Michael W Panas[3], David JP Ferguson[4], Marta Enciso[5], Matthew T O'Neill[1], Sash Lopaticki[1], Rebecca J Stewart[1,2], Grant Dewson[1,2], Gordon K Smyth[1,6], Brian J Smith[5], Seth L Masters[1,2], John C Boothroyd[3], Justin A Boddey[1,2*†], Christopher J Tonkin[1,2*†]

[1]The Walter and Eliza Hall Institute of Medical Research, Melbourne, Australia; [2]Department of Medical Biology, The University of Melbourne, Melbourne, Australia; [3]Department of Microbiology and Immunology, Stanford University School of Medicine, Stanford, United States; [4]Nuffield Department of Clinical Laboratory Science, Oxford University, John Radcliffe Hospital, Oxford, United Kingdom; [5]La Trobe Institute for Molecular Science, La Trobe University, Melbourne, Australia; [6]Department of Mathematics and Statistics, The University of Melbourne, Melbourne, Australia

*For correspondence: boddey@ wehi.edu.au (JAB); tonkin@wehi. edu.au (CJT)

†These authors contributed equally to this work

Competing interests: The authors declare that no competing interests exist.

**Abstract** Infection by *Toxoplasma gondii* leads to massive changes to the host cell. Here, we identify a novel host cell effector export pathway that requires the Golgi-resident aspartyl protease 5 (ASP5). We demonstrate that ASP5 cleaves a highly constrained amino acid motif that has similarity to the PEXEL-motif of *Plasmodium* parasites. We show that ASP5 matures substrates at both the N- and C-terminal ends of proteins and also controls trafficking of effectors without this motif. Furthermore, ASP5 controls establishment of the nanotubular network and is required for the efficient recruitment of host mitochondria to the vacuole. Assessment of host gene expression reveals that the ASP5-dependent pathway influences thousands of the transcriptional changes that *Toxoplasma* imparts on its host cell. All these changes result in attenuation of virulence of Δ*asp5* tachyzoites in vivo. This work characterizes the first identified machinery required for export of *Toxoplasma* effectors into the infected host cell.

## Introduction

The phylum Apicomplexa comprises a group of obligate intracellular parasites that cause a range of diseases by actively invading and replicating within host cells. Like all intracellular pathogens, these parasites extensively modify their host cells in order to prevent immune clearance, while permitting nutrient acquisition for growth. *Toxoplasma*, one of the most common human pathogens infecting 10–80% of individuals within a population, imparts a multitude of phenotypic changes on the infected host cell in order to promote survival and dissemination, including modulation of the inflammatory response (*Fischer et al. 1997*; *Braun et al., 2013*), hyper-migration of infected dendritic cells (*Lambert et al., 2006*), down-regulation of major histocompatibility complex (MHC) class II (*Lüder et al., 2003*), induction of c-Myc expression (*Franco et al., 2014*), activation of

**eLife digest** *Toxoplasma gondii* is a parasite that is thought to infect over two billion people worldwide. Often these infections cause no noticeable symptoms, but can cause serious illness in people with weakened immune systems. *Toxoplasma* parasites must enter human cells in order to survive. To dramatically increase their chances of survival, the parasites then deliver specialized proteins into the host cell that disarm the host's immune defenses. Understanding how these specialized proteins are transported from inside the parasite into the host cell, and how this process can be blocked, may lead to new treatments for these and related parasitic infections.

By genetically modifying *Toxoplasma* parasites to lack a parasite enzyme, Coffey et al. have now discovered that this molecule is required for correctly transporting parasite proteins. This enzyme is called aspartyl protease 5 (ASP5) and is found in the parasite in a structure called the Golgi apparatus, which acts as a main hub for protein transport.

ASP5 cuts proteins at a 'barcode' that is found in many different types of proteins, priming them for transport out of the parasite and for export into the host cell in some cases. Coffey et al. show that in parasites that lack ASP5, these proteins are no longer cleaved and are not transported correctly, blocking the activities that parasites normally perform to ensure their survival. Therefore, ASP5 plays an important role in transporting a wide range of proteins associated with disease, including transporting certain proteins directly into the host cell.

Future studies that compare parasites that lack ASP5 to normal parasites will aim to identify new proteins used by the parasites to defeat the host's immune defenses.

inflammasomes (*Ewald et al., 2014*), and recruitment of host endoplasmic reticulum (ER) (*Goldszmid et al., 2009*) and mitochondria (*Pernas et al., 2014*) to the parasitophorous vacuole membrane (PVM).

Over the last decade, the mechanisms of host cell modification by *Toxoplasma* have been explored. The first exported *Toxoplasma* effectors were identified through genetic quantitative trait loci mapping between progeny of crosses between virulent and avirulent strains. These proteins were shown to be protein kinases that are injected from the rhoptries into host cells during invasion (*Saeij et al., 2006*; *2007*; *Taylor et al., 2006*; *Peixoto et al., 2010*). Two canonical effector rhoptry proteins, ROP16 and ROP18, are only known to be injected into the host cell at the onset of invasion, where ROP16 levels peak within the host cell nucleus between 10 min and 4 hr post infection. ROP16 phosphorylates signal transducers and activators of transcription 1/3/5/6 (*Rosowski et al., 2012*; *Yamamoto et al., 2009*; *Jensen et al., 2013*; *Ong et al., 2010*), thus skewing the immediate-early immune response to limit parasite clearance (*Saeij et al., 2007*). While ROP16 and ROP18 were shown to be required for virulence differences between the three canonical *Toxoplasma* strains, they did not explain many other known phenotypic changes that occur during *Toxoplasma* infection of host cells.

Recently, an additional class of *Toxoplasma* effector proteins was identified as coming from the dense granules. These include dense granule protein 16 (GRA16), which is exported to the host cell nucleus post invasion via the dense granules, where it contributes to cell cycle arrest, potentially as a mechanism to prevent apoptosis (*Bougdour et al., 2013*). Other parasite processes and host pathways now known to be impacted by the GRA proteins include: a skewing of the immune response through the effector GRA24 (*Braun et al., 2013*), influencing nuclear factor kappa-light-chain-enhancer of activated B cells nuclear translocation in some strains via GRA15 (*Rosowski et al., 2011*), transport of small molecules across the PVM via GRA17 and GRA23 (*Gold et al., 2015*), generation of the nanotubular network (NTN, thought to aid nutrient acquisition [*Mercier, 2002*]) via GRA2 (and others) as well as recruitment of the host mitochondria to the PVM through the dense granule protein mitochondrial association factor 1 (MAF1) (*Pernas et al., 2014*). The recent and rapid discovery of these effectors suggests that there may be many more proteins that are exported via the dense granules and that they may use a conserved export pathway to mediate changes in the infected host cell.

While some exported proteins in *Toxoplasma* have been identified, there is currently little information about how these proteins are transported across the PVM and into the host cell. In the related malaria-causing parasites, *Plasmodium* spp., some of the mechanisms of protein export into the host erythrocyte have been revealed. Protein export by *P. falciparum* occurs almost immediately after invasion (*Riglar et al., 2013*), and cargo proteins traffic via the parasite's secretory pathway through the ER to the parasitophorous vacuole (PV) and across the PVM into the host cell (*Wickham, 2001*). In the majority of cases, a conserved pentameric motif, RxLxE/Q/D, referred to as the *Plasmodium* export element (PEXEL) or vacuolar transport signal (VTS), is required for export to the host cell (*Marti, 2004*, *Hiller, 2004*). In all published cases involving *Plasmodium* proteins, the PEXEL resides ~15–30 amino acids after the signal peptide (SP), where it acts as a proteolytic cleavage site (*Chang et al., 2008*; *Boddey et al., 2009*) for the ER-resident aspartyl protease plasmepsin V (PMV) (*Boddey et al., 2010*; *Russo et al., 2010*). PEXEL processing occurs after the leucine (RxL$^\downarrow$xE/Q/D), which reveals a new N-terminus that is acetylated in the ER (*Chang et al., 2008*; *Boddey et al., 2009*). The current hypothesis is that the exposed new N-terminus ($^{Ac-}$xE/Q/D) permits cargo selection for targeting to a parasite translocon located at the PVM, known as PTEX (for *Plasmodium* translocon of exported proteins) (*de Koning-Ward et al., 2009*; *Elsworth et al., 2014*; *Beck et al., 2014*). Effectors must be unfolded for translocation (*Gehde et al., 2009*) through PTEX into the host cell before refolding and trafficking to their final destination in the host cell.

Given that several dense granule proteins are exported by *Toxoplasma*, we investigated whether a conserved pathway is used and whether it shares any similarities with the *Plasmodium* system. Here, we identify the novel Golgi-resident aspartyl protease 5 (ASP5) that is the first known component of the dense granule export machinery in *Toxoplasma*. Our study of ASP5 has revealed a novel mechanism of protein export in this parasite and extended our understanding of the importance of this pathway in inducing changes to the infected host cell. This work highlights similarities and important differences between mechanisms of protein export in the agriculturally and medically important Apicomplexan phylum.

## Results

### A pentameric motif is necessary for proteolytic processing of GRA16 and export to the host cell

Several hundred *P. falciparum* proteins contain a pentameric amino acid motif, also known as the PEXEL, that is essential for export into the infected erythrocyte (*Marti, 2004*; *Hiller, 2004*). Within the N-terminus of GRA16 (*Bougdour et al., 2013*), we observed a PEXEL-like motif (RRLAE) after the SP, at amino acid positions 63 to 67 (*Figure 1A*). To determine whether the PEXEL-like motif was involved in protein trafficking in *Toxoplasma*, we undertook a mutational analysis of GRA16 at the endogenous locus. This was achieved through double-homologous recombination whereby the endogenous *GRA16* gene was replaced with either wild-type (WT) *gra16* encoding the native PEXEL-like motif RRLAE and fused to a C-terminal hemagglutinin (HA) tag (GRA16$_{WT}$-HA), or a version of *gra16* with its PEXEL-like motif mutated from RRLAE to AAAAE (GRA16$_{AAAAE}$-HA). The resulting lines were analyzed for proteolytic processing and trafficking. Immunoblot analysis showed that GRA16$_{WT}$-HA is represented by a strong signal at ~57 kDa and two minor species at approximately 60 kDa and 54 kDa, respectively (*Figure 1B*). Following mutation of the PEXEL-like motif (GRA16$_{AAAAE}$-HA), the two lower molecular weight species were not observed, demonstrating that the mutated protein was no longer processed in the same way. The result is consistent with the slowest migrating species representing signal peptidase-cleaved GRA16, while the size shift of the dominant ~57 kDa species is consistent with cleavage of the PEXEL-like motif located ~45 residues beyond the SP. Interestingly, while both proteins were expressed from the endogenous locus, the amount of GRA16$_{AAAAE}$-HA protein was dramatically reduced (*Figure 1B*), suggesting the mutant protein was degraded in the absence of appropriate N-terminal processing. These results are consistent with the PEXEL-like motif being a proteolytic cleavage site similar to that observed in *Plasmodium* spp. (*Boddey et al., 2010*; *Russo et al., 2010*).

We next sought to determine whether the PEXEL-like motif was required for GRA16 trafficking to the host cell, as is true for the PEXEL motif in *Plasmodium* spp. (*Hiller, 2004*, *Marti, 2004*). Human foreskin fibroblasts (HFFs) were infected with parasites expressing GRA16$_{WT}$-HA and GRA16$_{AAAAE}$-

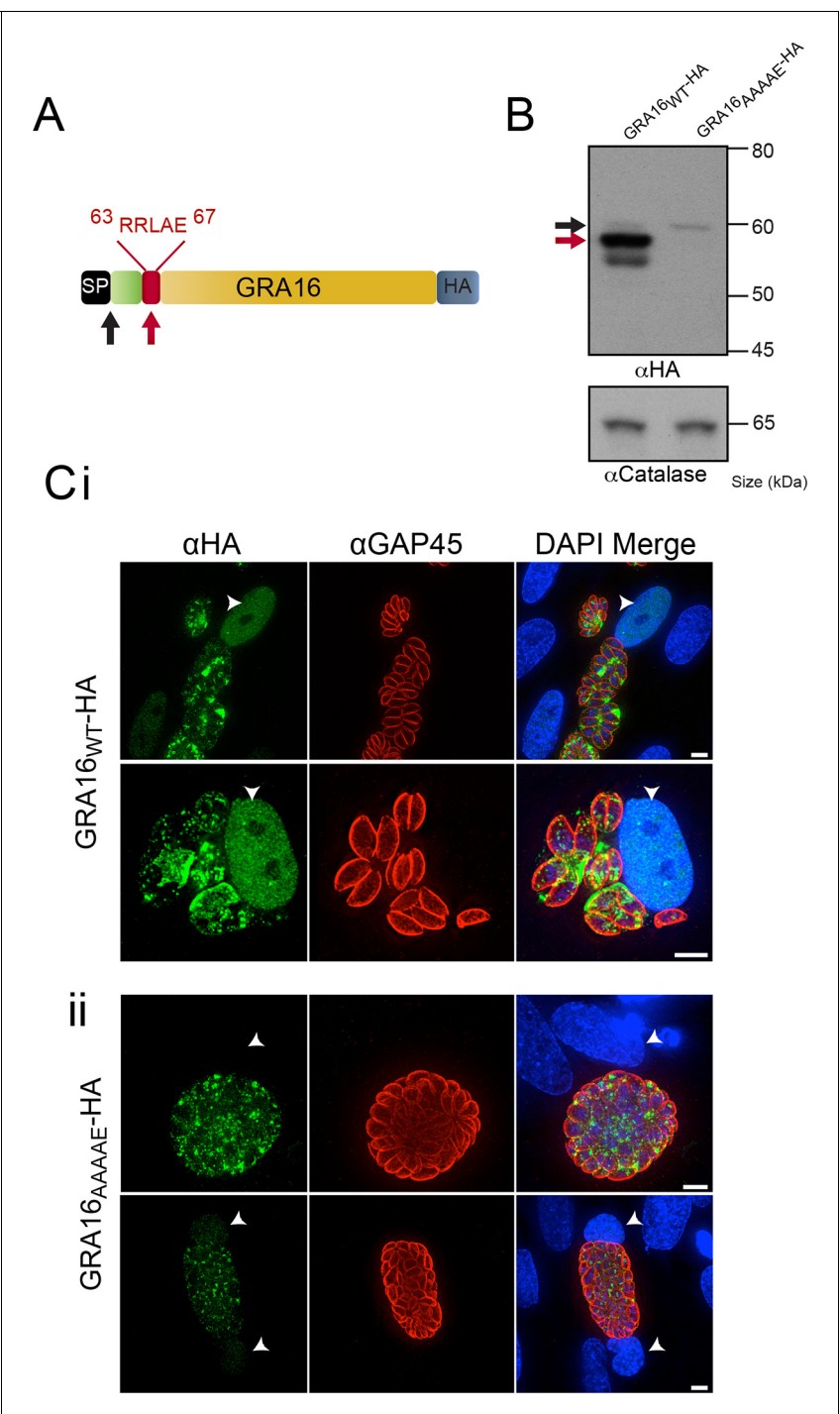

**Figure 1.** GRA16 contains a PEXEL-motif that is required for processing and export. (A) Schematic representation of GRA16 containing an N-terminal SP for entry into the secretory pathway and a PEXEL-like (TEXEL) motif RRLAE found at residues 63–67. Arrows relate to predicted sizes of bands seen by Western blot. (B) Western blot of GRA16$_{WT}$-HA and GRA16$_{AAAAE}$-HA. GRA16$_{WT}$-HA has three molecular weight species, the uppermost (black arrow) being consistent with SP cleaved, the middle (red arrow) consistent with TEXEL cleavage and the lowest band, which is a potential degradation product. GRA16$_{AAAAE}$-HA is present as only the slowest migrating species, consistent with the expected size of signal peptide cleaved, TEXEL uncleaved. αCatalase antibodies are used as a loading control. (C) Localization of GRA16$_{WT}$-HA and GRA16$_{AAAAE}$-HA. (i) As previously reported, GRA16$_{WT}$-HA is exported into the host cell where it accumulates in the nucleus (arrowheads) while also being present within tachyzoites and the PV space. (ii) GRA16$_{AAAAE}$-HA is exported far less efficiently (a small amount can be observed in the host cell nucleus) while the majority of this protein accumulates within tachyzoites and the PV space. Scale bar is 5 μm. HA, hemagglutinin; PEXEL, *Plasmodium* export element; PV, parasitophorous vacuole; SP, signal peptide; TEXEL, *Toxoplasma* export element.

HA for 24 hr and the localization of the proteins was determined by immunofluorescence assay (IFA) using anti-HA antibodies. GRA16$_{WT}$-HA was observed within the host cell nucleus, as previously reported (*Bougdour et al., 2013*), as well as at the PV and within parasites (*Figure 1C-i*). In contrast, GRA16$_{AAAAE}$-HA was observed either within parasites, in small punctate structures reminiscent of the Golgi or in the PV space between parasites (*Figure 1C-ii*). In a minority of cells, a small amount of exported GRA16$_{AAAAE}$-HA could be observed within the host nuclei (*Figure 1C-ii*, panel 2); how-ever, there was a large and clear defect in export in the mutant line. Taken together, this demon-strates that the PEXEL-like motif is required for correct proteolytic processing of GRA16 and efficient export to the host cell. We therefore termed this motif the *Toxoplasma* export element (TEXEL).

## ASP5 cleaves the TEXEL motif and is inhibited by a TEXEL mimetic inhibitor

In *Plasmodium* spp., the PEXEL is cleaved by the ER-resident aspartyl protease plasmepsin V (PMV [*Boddey et al., 2010*; *Russo et al., 2010*]). We hypothesized that an orthologous protease in *Toxo-plasma* is required for cleavage of the TEXEL in GRA16 and potentially other *Toxoplasma* proteins. We searched ToxoDB (http://toxodb.org) using *P. falciparum* PMV (PfPMV) as a query and the top Basic Local Alignment Search Tool (BLAST) hit was aspartyl protease 5 (ASP5, TGME49_242720), consistent with previous phylogenetic analysis of this group of proteases in Apicomplexa (*Shea et al., 2007*). An alignment of the two proteins revealed that they share approximately 33% similarity and 14% identity across the full-length alignment (*Figure 2—figure supplement 1*). The proteins shared several key features, including an N-terminal SP, a core aspartyl protease domain (with DTG and DSG residues defining the catalytic dyad), a plant-like nepenthesin fold, as well as a C-terminal transmembrane domain (*Hodder et al., 2015*). While ASP5 contains a significantly longer SP and C-terminal tail sequence than PfPMV, it lacks the helix-turn-helix motif found in PMV from all *Plasmodium* spp. that is hypothesized to interact with other ER proteins (*Hodder et al., 2015*).

To characterize ASP5 within parasites, we tagged the 3' end of the endogenous gene with a tri-ple-hemagglutinin (HA$_3$) tag in the RHΔ*ku80* background (*Huynh and Carruthers, 2009*). Immuno-blot analysis with anti-HA antibodies revealed ASP5$_{WT}$-HA$_3$ is present as two major species of approximately 90 and 55 kDa (*Figure 2A*), consistent with a signal peptidase-cleaved species and possibly an activated form, respectively (*Figure 2A*). A mutant form of ASP5, where the conserved aspartic acid catalytic residues were mutated to alanine (ASP5$_{D431A, D682A}$-HA$_3$; herein referred to as ASP5$_{MUT}$-HA$_3$), was observed predominantly as the ~90 kDa form, suggesting that ASP5 may auto-activate to produce the ~55 kDa species (*Figure 2A*, *Figure 2—figure supplement 2*). ASP5 was previously localized to the Golgi when tagged with a Ty1 epitope tag (*Shea et al., 2007*). Using immunofluorescence microscopy with anti-HA antibodies, we observed ASP5$_{WT}$-HA$_3$ in apical puncta that co-localized with GalNAc-YFP, a known Golgi marker (*Nishi et al., 2008*) (*Figure 2B*). ASP5$_{MUT}$-HA$_3$ also localized to discrete puncta adjacent to the nucleus, representative of the Golgi (*Figure 2B*).

To determine whether ASP5 could cleave the TEXEL motif, we immunopurified ASP5$_{WT}$-HA$_3$ from transgenic parasites using anti-HA agarose and incubated it with a fluorogenic peptide containing the TEXEL sequence RRLAE from GRA16, as previously performed for PMV with PEXEL substrates (*Boddey et al., 2010*; *Russo et al., 2010*). ASP5$_{WT}$-HA$_3$ efficiently cleaved the GRA16 peptide with $K_m$ 47.8 ± 18.4 μM (mean ± SD) (*Figure 2C* and *Figure 2—figure supplement 3C*). However, only minimal cleavage was observed when the TEXEL peptide was mutated from RRLAE to AAAAE (*Figure 2C*), similar to the specificity observed for PMV (*Boddey et al., 2010*; *Russo et al., 2010*). ASP5$_{WT}$-HA$_3$ activity was optimal at pH 5.5 (*Figure 2—figure supplement 3A*), in contrast to pH 6.4 for PMV (*Boddey et al., 2010*; *Russo et al., 2010*), consistent with the Golgi being a more acidic environment than the ER (*Wu et al., 2000*). To control against proteolysis by potentially contaminat-ing enzymes in the WT ASP5 preparation, we immunopurified ectopic ASP5$_{MUT}$-HA$_3$ from otherwise WT parasites, as above, and incubated it with the GRA16 TEXEL peptides. No cleavage was observed (*Figure 2C*), demonstrating that the GRA16 TEXEL peptide is specifically cleaved by ASP5 and that this is dependent on the catalytic residues D431 and D682.

To determine the amino acid position of substrate processing by ASP5, we used liquid chroma-tography (LC) combined with tandem mass spectrometry (MS/MS) to examine the GRA16 peptide cleavage products (*Figure 2D*). Peptides incubated in buffer alone remained intact in contrast to

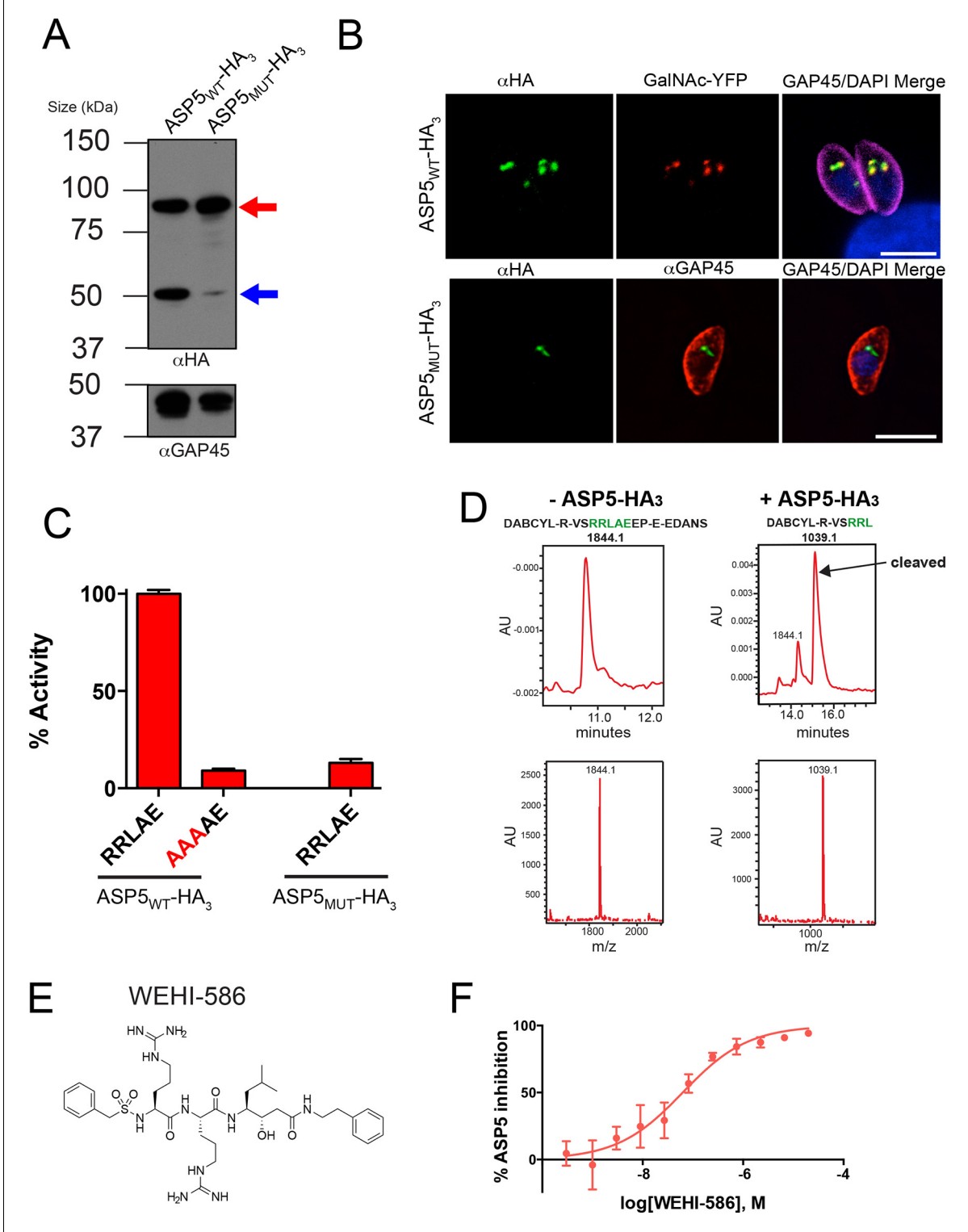

**Figure 2.** ASP5 specifically cleaves the GRA16 TEXEL. (**A**) Western blot of endogenously tagged ASP5 (ASP5$_{WT}$-HA$_3$) and ectopic ASP5$_{D431A, D682A}$-HA$_3$ (ASP5$_{MUT}$-HA$_3$) in parasites shows two predominant species. The upper band (red arrow) is consistent with a signal peptidase-cleaved species and the lower (blue arrow) may be auto-activation, as it is greatly diminished for ASP5$_{MUT}$-HA$_3$. αGAP45 antibodies are used as a loading control. (**B**) Endogenously-expressed ASP5$_{WT}$-HA$_3$ co-localizes with the Golgi marker GalNAc-YFP (upper panel) and this localization is unaffected for the catalytic mutant enzyme (ASP5$_{MUT}$-HA$_3$) (lower panel). (**C**) Immunoprecipitated ASP5$_{WT}$-HA$_3$, but not ASP5$_{MUT}$-HA$_3$, cleaves GRA16 TEXEL (DABCYL-R-VS**RRLAE**EP-E-EDANS) but not the RRL>AAA peptide. (**D**) LC chromatogram (214 nm) of the fluorogenic GRA16 TEXEL peptide (upper left)

*Figure 2 continued on next page*

*Figure 2 continued*

incubated in buffer alone (-ASP5-HA$_3$) with MS analysis showing the parent ion of the unprocessed fluorogenic peptide (lower left). LC chromatogram (214 nm) of the fluorogenic GRA16 TEXEL peptide after incubation at 37°C for 48 hr with ASP5 ( +ASP5$_{WT}$-HA$_3$) (upper right), showing the N-terminal product of processing within the TEXEL after leucine (DABCYL-R-VS**RRL**) at 15.5 min while the remaining unprocessed fluorogenic peptide is observed at 14.3 min. MS analysis showing the parent ion of the processed N-terminal cleavage product DABCYL-VS**RRL** (lower right). (**E**) Structure of WEHI-586 (RRL$_{Statine}$). (**F**) Dose response curve showing inhibition of ASP5$_{WT}$-HA$_3$ activity by WEHI-586 with IC$_{50}$ of 63 ± 15 nM. Data shown are mean ± standard deviation from three experiments. Scale bar is 5 μm. ASP5, Aspartyl Protease 5; HA$_3$, triple-hemagglutinin; LC; liquid chromatography, MS; mass spectrometry.

The following figure supplements are available for figure 2:

**Figure supplement 1.** Alignment of PfPMV and TgASP5 sequences.

**Figure supplement 2.** ASP5 may undergo auto proteolysis.

**Figure supplement 3.** Enzymological Characterization of ASP5.

**Figure supplement 4.** Synthesis scheme for generation of WEHI-586.

peptides incubated with ASP5$_{WT}$-HA$_3$, which resulted in the generation of a product corresponding to processing within the TEXEL after leucine (DABCYL-R-VS**RRL**↓). This processing event after the leucine residue, hereafter referred to as the P$_1$ position, is the identical site of processing of the PEXEL by PMV in both *P. falciparum* and *P. vivax* (*Boddey et al., 2009*; *2010*; *Russo et al., 2010*; *Sleebs et al., 2014b*).

To further examine the specificity of ASP5 for the TEXEL sequence, we designed a peptide-like inhibitor that directly mimics the TEXEL sequence RRLAE from GRA16 but that contains the non-cleavable amino acid, statine (RRL$_{Statine}$; WEHI-586, *Figure 2E*). This compound is predicted to bind the active site of ASP5 and mimic the transition state of GRA16 TEXEL cleavage, thus inhibiting the enzyme. Incubation of ASP5$_{WT}$-HA$_3$ with WEHI-586 blocked cleavage of the GRA16 peptide with IC$_{50}$ of 63 ± 15 nM (mean ± standard error of the mean) (*Figure 2F*), demonstrating the potent affinity of the TEXEL sequence for ASP5. Taken together, these results demonstrate that ASP5 is a Golgi-resident protease that cleaves the GRA16 TEXEL motif after the leucine residue and can be potently inhibited by a TEXEL-mimetic small molecule.

## ASP5 has specific and unique substrate specificity

To investigate the substrate selectivity of ASP5 and directly compare it with PMV, we incubated ASP5$_{WT}$-HA$_3$ and PfPMV-HA with peptides containing different point mutations at the TEXEL and PEXEL motifs, based on RRLAE from GRA16 and RTLAQ from the *P. falciparum* exported protein, knob associated histidine rich protein (KAHRP), respectively. PfPMV-HA behaved as expected (*Boddey et al., 2010*; *2013*), cleaving peptides containing the WT KAHRP PEXEL but not P$_3$ (R>K) or P$_1$ (L>I) point mutations (*Figure 3A-i*). This *Plasmodium* enzyme also cleaved the peptide containing the WT GRA16 TEXEL, with notably higher efficiency than it cleaved the peptide KAHRP; however, it did not cleave GRA16 peptides containing TEXEL mutations at P$_3$ (R>A) or P$_1$ (L>A), as expected based on the known specificity of this protease (*Figure 3A-i*). Similarly, ASP5 cleaved the GRA16 TEXEL peptide but did not cleave peptides containing mutations of the TEXEL, P$_3$ (R>A) or P$_1$ (L>A), demonstrating that the P$_3$ and P$_1$ positions of the substrate (i.e. arginine and leucine, respectively) are important for ASP5 activity, as is the case for PMV (*Figure 3A-ii*) (*Boddey et al., 2010*; *Russo et al., 2010*; *Sleebs et al., 2014b*). In contrast to PMV, ASP5 did not cleave KAHRP peptides above background levels (*Figure 3A-ii*). Replacement of the P$_2$ residue in the KAHRP PEXEL (threonine) with the corresponding residue in GRA16 (arginine) (i.e. P$_2$ T>R) resulted in a 3-fold increase in cleavage, demonstrating the importance of the P$_2$ position for ASP5 activity, although cleavage of this peptide was still well below that seen for the native GRA16 TEXEL peptide (*Figure 3A-ii*). This demonstrates that while ASP5 and PMV both cleave peptides containing RxL sequences, they do not share identical substrate specificity. To further investigate the specificity of ASP5, point mutations were introduced at different positions of the GRA16 TEXEL substrate. This

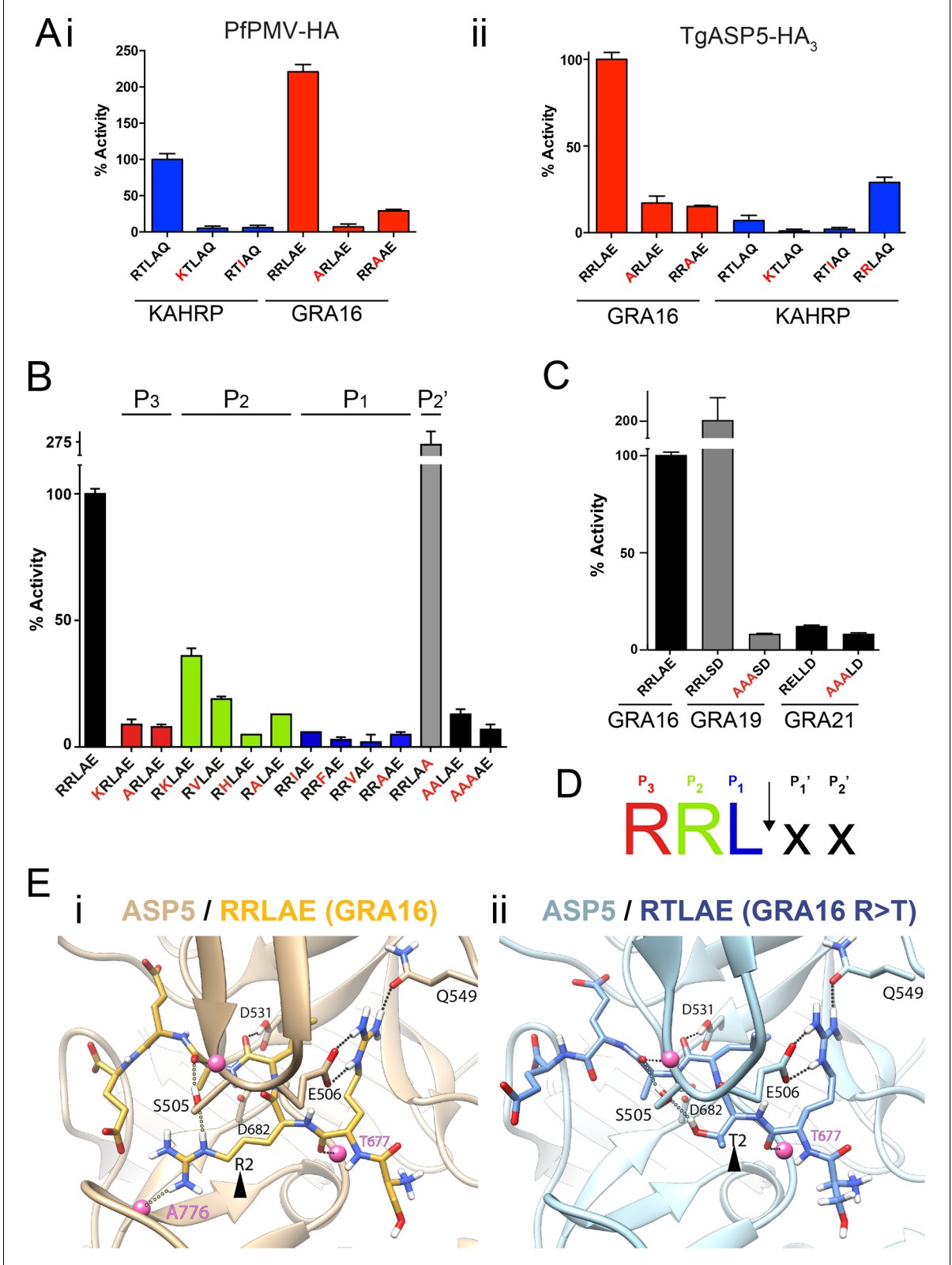

**Figure 3.** ASP5 is highly selective for 'RRL' substrates. (**A**) (**i**) Activity of immunoprecipitated PfPMV-HA against KAHRP- and GRA16-based fluorogenic DABCYL/EDANS peptides. PfPMV-HA is able to cleave peptides containing KAHRP PEXEL and GRA16 TEXEL sequences but not corresponding mutants (red amino acids). Note the GRA16 'RRLAE' TEXEL is cleaved approximately twice as efficiently as the KAHRP 'RTLAQ' PEXEL. (**ii**) Cleavage of

*Figure 3 continued on next page*

*Figure 3 continued*

substrates by immunoprecipitated TgASP5-HA$_3$, as in (i). ASP5 cleaves the wild type GRA16 TEXEL but is unable to efficiently process the corresponding point mutants in GRA16 or any KAHRP peptides. Mutation of the P$_2$ threonine in KAHRP for arginine (T>R) marginally increases processing. (B) Substrate specificity of ASP5-HA$_3$ in relation to the P$_1$, P$_2$, P$_3$ and P$_2$' positions. This protease is unable to tolerate conservative and non-conservative changes at P$_1$, P$_2$ or P$_3$; however, this constriction appears to be more relaxed at P$_2$'. (C) ASP5$_{WT}$-HA$_3$ cleaves the GRA16 TEXEL, as well as the TEXEL from the dense granule protein GRA19, but not a similar motif in GRA21, or peptides containing RRL>AAA mutations. (D) Preferred TEXEL consensus with the position of cleavage by ASP5 indicated (arrow), color-coded according to (B). (E) (i) Structural model of ASP5 in complex with the TEXEL from GRA16 (SRRLAEE) colored gold; or (ii) with a point mutant of GRA16 containing threonine at P$_2$ (SRTLAEE) colored blue to explain why arginine is preferred at P$_2$. Arrowheads denote the P$_2$ position in each substrate. Heteroatoms are colored white: hydrogen, blue: nitrogen and red: oxygen. Several backbone groups in ASP5 are highlighted as pink spheres. Hydrogen bonds between the GRA16 peptides and ASP5 are shown as dotted lines; colored lines highlight the hydrogen bond interactions that differ between the two substrates. ASP5, Aspartyl Protease 5; HA, hemagglutinin; KAHRP, knob associated histidine rich protein; PEXEL, plasmodium export element; PfPMV, *P. falciparum* PMV; TEXEL, *Toxoplasma* export element.

demonstrated that ASP5 does not well tolerate conservative and non-conservative changes at the P$_1$, P$_2$ or P$_3$ positions. It appeared that ASP5 could cleave RKL at ~35–40% of WT, yet the physiological relevance of this is not known (*Figure 3B*). Interestingly, mutation of the GRA16 TEXEL P$_2$' residue E>A resulted in enhanced processing, illustrating that this position in ASP5 substrates may not be essential for activity, similar to PMV, but can alter cleavage efficiency (*Boddey et al., 2009*; *2010*; *Sleebs et al., 2014a*; *2014b*).

PEXEL-like sequences have previously been identified in GRA19 (RRLSD) and GRA21 (RRLAE and RELLD) (*Hsiao et al., 2013*). To examine whether ASP5 can cleave these sequences, peptides were synthesized containing RRLSD (GRA19) and RELLD from GRA21. These peptides were incubated with ASP5$_{WT}$-HA$_3$, as above, alongside corresponding RRL>AAA or REL>AAA mutants (*Figure 3C*). The WT GRA19 TEXEL peptide was processed efficiently and this was inhibited when the TEXEL was mutated from RRL to AAA (*Figure 3C*). In contrast, the RELLD sequence in GRA21, and the corresponding AAALD mutant, were not processed by ASP5 (*Figure 3C*). Since the RRLAE in the N-terminus of GRA21 is the same sequence as the TEXEL in GRA16, it is highly likely that ASP5 processes GRA21 at this position. Taken together, this work demonstrates that ASP5 has relatively strict requirement for arginine at P$_3$ and P$_2$, and leucine at P$_1$ of its substrates (*Figure 3D*), and that residues at P$_1$' and P$_2$' are dispensable for processing but can influence the efficiency of cleavage by this enzyme.

## ASP5 modeling reveals key interactions with the TEXEL of GRA16

To investigate the structural basis for substrate selection by ASP5, we modeled the tertiary structure of this enzyme bound to the GRA16 substrate using the crystal structure of PMV from *P. vivax* complexed with the PEXEL mimetic inhibitor WEHI-842 (*Hodder et al., 2015*) and *P. vivax* plasmepsin IV in complex with Pepstatin A (*Bernstein et al., 2003*) as templates (*Figure 3E*). The model shows that the guanidyl side-chain of arginine at P$_3$ (the first position in the TEXEL sequence) forms interactions with the side-chains of E506 and Q549 in a manner completely analogous to that observed in the structure of PMV in complex with the statine inhibitor, WEHI-842 (*Hodder et al., 2015*) (*Figure 3E-i*). Furthermore, the leucine at P$_1$ of the TEXEL is surrounded by hydrophobic residues I429, Y503, F546 and I554 of ASP5; the isoleucine at position 554 in ASP5 is a valine in PMV, while the other residues, I, Y, F are identical between ASP5 and PMV (*Figure 3E-i*). Our TEXEL cleavage data described above revealed that, unlike PMV, arginine is strongly preferred at the P$_2$ position for ASP5 activity and that this *Toxoplasma* enzyme could not efficiently process the PEXEL motif from KAHRP, which contains threonine at P$_2$. This was supported using our model, as the AutoDock potential predicted that ASP5 binds the GRA16 peptide (SRRLAEE) 5 kJ/mol more tightly than an SR**T**LAEE mutant form of GRA16 (*Figure 3E-ii*). In this mutant substrate, the side-chain guanidine of arginine at P$_3$ is still clamped by the side-chain carboxylate and amide of ASP5 residues E506 and Q549, respectively, and the backbone carbonyl oxygen of the arginine residue also forms a hydrogen bond with the backbone amide of T677 of ASP5. The backbone carbonyl of leucine at P$_1$ in the GRA16 TEXEL forms a hydrogen bond with the side-chain hydroxyl and the backbone amide of S505. The guanidine side-chain of arginine at P$_2$ of native GRA16 forms hydrogen bonds with the side-chain hydroxyl of S505 and the backbone carbonyl of A776 (*Figure 3E-i*), whereas the mutated

GRA16 substrate containing threonine at $P_2$ forms only a single hydrogen bond with the side-chain hydroxyl of S505 (*Figure 3E-ii*). Taken together, these differences in binding interactions accounts for ~50% of the total difference in the calculated binding affinity between the two substrates, providing a clear structural explanation for the substrate specificity (i.e. RRL) observed for ASP5.

We also used structural modeling to understand the substrate preference at other sites within the TEXEL motif. Mutation of leucine at $P_1$ of the GRA16 TEXEL to valine reduces the calculated binding energy by 6 kJ/mol, which is in line with our observations that mutations at this position significantly reduce ASP5 activity. The major source of the reduction in binding energy in the L>V mutation arises from a reduction in electrostatic interaction, similar to that seen for the R>T mutation above. Mutation of alanine at $P_1'$ (position 4 of the GRA16 TEXEL, RRLAE) to valine is predicted to slightly increase the binding energy, but by less than 1 kJ/mol. The small change in calculated binding energy is consistent with a lack of sensitivity at this position in the TEXEL sequence recognized by ASP5. Interestingly, mutation of glutamine at $P_2'$ (position 5 of the GRA16 TEXEL) to alanine causes an 11 kJ/mol reduction in calculated binding affinity in the model, in contrast to the increase in ASP5 activity observed in vitro (*Figure 3B*). It is possible the glutamine reside at position 6 (i.e. RRLAE**E**) can act as a surrogate for the loss of glutamine at position 5 in this interaction (*Figure 3E-ii*).

## Deletion of ASP5 causes loss of fitness and the inability of parasites to process GRA16

Following validation of ASP5 as the TEXEL-cleaving protease in vitro, we sought to determine whether this occurs in parasites in vivo through deletion of the *ASP5* gene in parasites expressing GRA16-HA. Utilizing a double homologous recombination strategy combined with an *ASP5*-targeted CRISPR approach, we were able to successfully disrupt the *ASP5* gene, where the 3' flank underwent homologous recombination, while apparent lack of *Not*I cleavage and the presence of a Cas9-induced cut site resulted in the whole plasmid integrating non-homologously at that site, meaning that a green fluorescent protein (GFP) expression cassette also integrated (*Figure 4—figure supplement 1A-i*). This integration was confirmed through polymerase chain reaction (PCR) and sequencing of the *ASP5* locus (*Figure 4—figure supplement 1A-ii* and data not shown). To determine overall qualitative changes in asexual growth rate, WT and Δ*asp5* tachyzoites were grown for 7 days in a plaque assay (*Figure 4A-i*) and we observed that the plaques of the Δ*asp5* parasites were smaller than those generated by WT parasites, demonstrating that Δ*asp5* parasites have a clear growth disadvantage under simple in vitro growth conditions. We subsequently generated a second Δ*asp5* mutant in the RHΔ*hxgprt* background using CRISPR/Cas9 to yield Δ*asp5*<sub>CRISPR</sub> (*Figure 4—figure supplement 1B*), which had a similar growth defect to the Δ*asp5*:GRA16-HA parasites. This defect was restored following complementation with a stably-integrated copy of *ASP*5 (Δ*asp5*<sub>CRISPR</sub>: ASP5<sub>WT</sub>-HA<sub>3</sub>) driven from the tubulin promoter (*Figure 4A-ii*.)

To assess whether the loss of ASP5 resulted in a reduced intracellular growth rate, we assessed replication of parasites 16 hr after infection (*Figure 4B*). From this analysis, it is clear that Δ*asp5*<sub>CRISPR</sub> tachyzoites have no major difference in intracellular replication to either WT or Δ*asp5*<sub>CRISPR</sub>:ASP5<sub>WT</sub>-HA<sub>3</sub> parasites. This suggests that smaller plaque size of Δ*asp5*<sub>CRISPR</sub> tachyzoites, compared to WT and Δ*asp5*<sub>CRISPR</sub>:ASP5<sub>WT</sub>-HA<sub>3</sub> parasites, is not due to retardation in replication.

To assess whether ASP5 is the TEXEL cleaving protease in tachyzoites, we examined the processing and trafficking of GRA16-HA in the presence and absence of ASP5. Western blot analysis of GRA16-HA in otherwise WT parasites yielded the same three bands as seen in *Figure 1B*, consistent with cleavage within the TEXEL motif, whereas in the Δ*asp5* parasites, GRA16-HA migrated as a larger protein that had not been processed correctly, mirroring the GRA16<sub>AAAAE</sub>-HA profile (*Figure 4C*, *Figure 1B*). The localization of GRA16-HA was then investigated by IFA in the presence and absence of ASP5. While GRA16-HA produced by WT parasites was observed in the host nucleus as expected, this effector was no longer exported into the host cell during infection with Δ*asp5* parasites and instead appeared to localize to an internal structure reminiscent of the Golgi and in the PV space (*Figure 4D*), similar to the GRA16<sub>AAAAE</sub>-HA mutant (*Figure 1C-ii*). This confirms that processing by ASP5 is essential for correct trafficking of GRA16 from the parasite into the infected host cell.

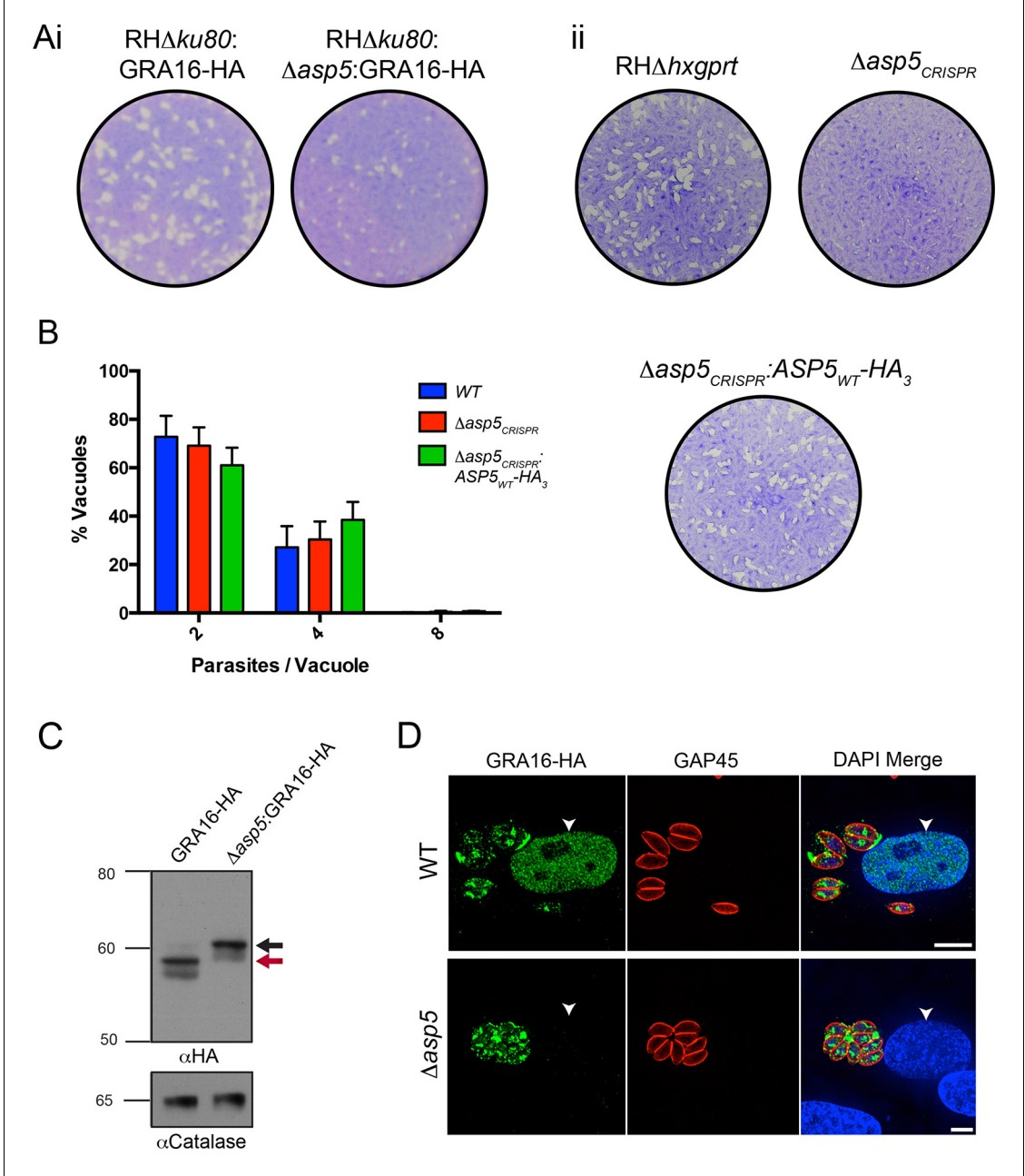

**Figure 4.** ASP5 is required for cleavage and export of GRA16. (**A**) (i) A plaque assay on confluent HFF monolayers, stained with crystal violet at 7 days post infection where plaques produced by Δku80:GRA16-HA:Δasp5 parasites are smaller than those made by WT (Δku80:GRA16-HA) parasites. (ii) As in (i), where the plaques formed by Δasp5$_{CRISPR}$ parasites are diminished in comparison to parental wildtype (RHΔhxgprt) and Δasp5$_{CRISPR}$:ASP5$_{WT}$-HA$_3$ parasites. (**B**) Replication assay. Tachyzoites were grown in HFFs and fixed at 16 hr post infection. Samples were stained with αGAP45 antibodies and counted. n = 3 independent experiments where > 50 vacuoles were counted, values are mean ± standard error of the mean. (**C**) Western blot of GRA16-HA in Δku80:GRA16-HA (lane 1) and Δku80:GRA16-HA:Δasp5 (lane 2) parasites. The black arrow corresponds to the predicted signal peptidase cleaved-species and the red arrow to the TEXEL cleaved product, as outlined in **Figure 1A**. Catalase antibodies are used as a loading control. (**D**) IFA showing GRA16-HA is exported into the host cell nucleus in otherwise WT parasites (top panel) but not in Δasp5 parasites (GFP-positive, signal diminished in comparison to the strong GRA16-HA in the 488 nm channel). White arrowheads indicate host nuclei. Scale bar is 5 μm. GFP, green fluorescent protein; HA, hemagglutinin; HFF, human foreskin fibroblasts; IFA, immunofluorescence assay; WT, wild type.

The following figure supplement is available for figure 4:

**Figure supplement 1.** Generation and complementation of Δasp5 *parasites*.

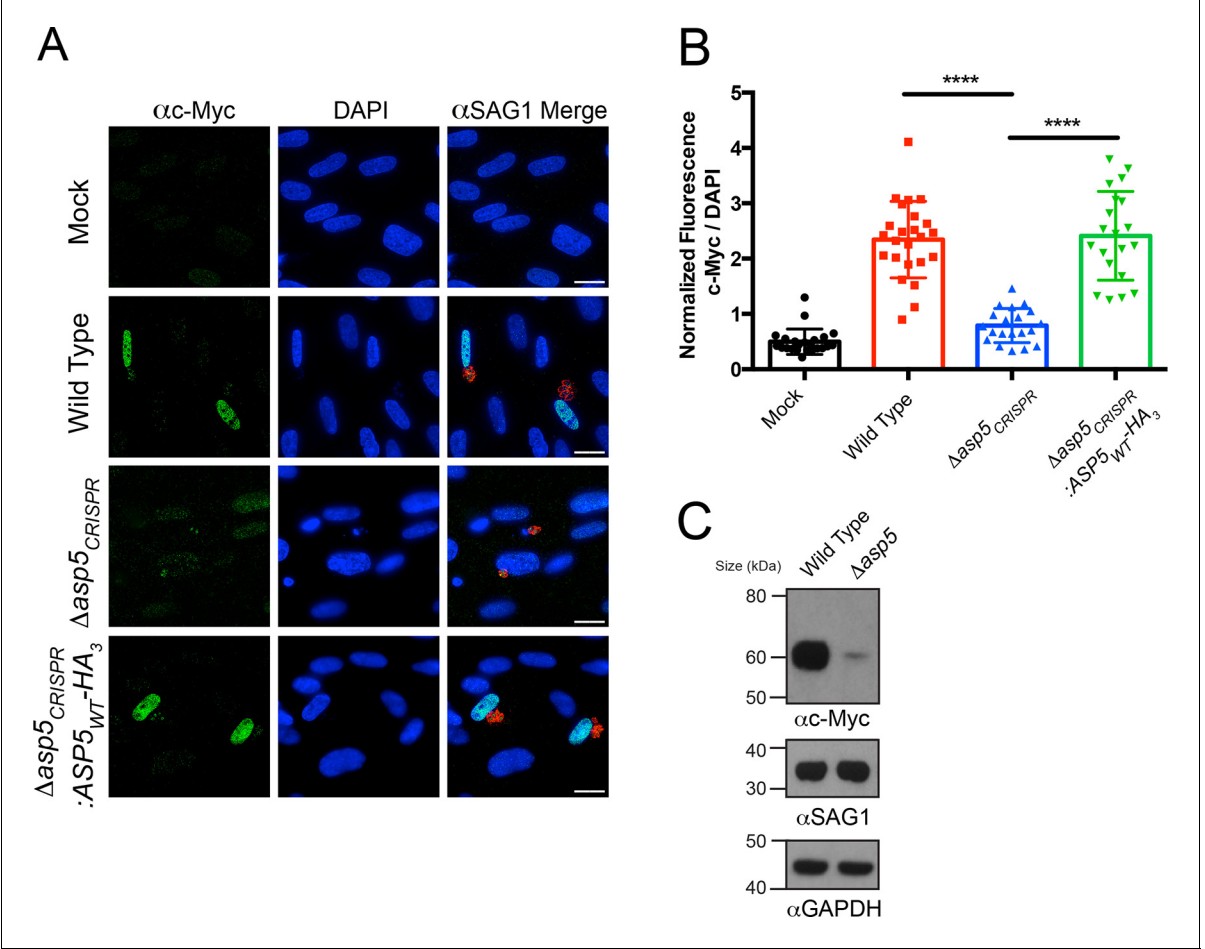

**Figure 5.** Induction of host c-Myc is ASP5-dependent. (**A**) Representative IFAs 14 hr after infection of c-Myc expression in confluent HFFs. Mock-infected cells express very little c-Myc while infection with WT parasites leads to a dramatic up-regulation of this transcription factor. Δ*asp5_CRISPR*-infected cells express marginally more c-Myc than mock-infected, which is complemented by the re-introduction of ASP5 (Δ*asp5_CRISPR*:ASP5_WT-HA_3). (**B**) Quantitation of c-Myc signal (as a ratio of DAPI signal) in cells from (A), *P* = 0.0001, values are mean ± standard deviation, unpaired t-test, n ≥ 20 nuclei from cells infected with single vacuoles. (**C**) Western blot showing up-regulation of c-Myc upon wild type infection, which is drastically decreased following deletion of *ASP5*. αSAG1 and αGAPDH serve as parasite and host loading controls, respectively. Scale bars are 20 μm. HA_3, triple-hemagglutinin; HFFs, human foreskin fibroblasts; IFA, immunofluorescence assay; WT, wild type.

## ASP5 is required for the activation of c-Myc during infection

We sought to determine the importance of ASP5 in controlling other cellular phenotypes that *Toxoplasma* imparts on its host cell. It has recently been shown that *Toxoplasma* tachyzoites, but not the related *Neospora* species, actively induce expression of host c-Myc following infection (*Franco et al., 2014*). This activation of c-Myc was not induced in response to parasite invasion or injection of rhoptry proteins (*Franco et al., 2014*), suggesting that one or more dense granule proteins may be responsible. To determine whether ASP5 is involved in up-regulation of host c-Myc, HFFs were infected with WT, Δ*asp5* or Δ*asp5_CRISPR*:ASP5_WT-HA_3 parasites, and c-Myc expression was measured by IFA and immunoblot. While uninfected HFFs showed little c-Myc expression by IFA, cells infected with WT *Toxoplasma*, or parasites with complemented ASP5 expression, had almost universal induction of c-Myc in their nuclei, as previously reported (*Figure 5A*) (*Franco et al., 2014*). Upon infection with Δ*asp5* parasites, a sharp reduction in c-Myc expression within the nuclei was observed (*Figure 5A*). Quantification by IFA showed that HFFs infected with parasites lacking ASP5 expressed approximately 6.4-fold less c-Myc than those infected with WT parasites (normalized ratio of c-Myc to 4′,6-diamidino-2-phenylindole [DAPI]) (*Figure 5B*). To confirm this, c-Myc induction was measured by immunoblot of whole cell protein fractions. While c-Myc expression was

induced in host cells infected with WT parasites, the signal was dramatically reduced in HFFs infected with the same number of Δasp5 parasites (*Figure 5C*), confirming that ASP5 is required for *Toxoplasma* to induce c-Myc in infected cells. Together, this work suggests that the up-regulation of c-Myc induced by tachyzoites is controlled by one or more ASP5-dependent proteins.

Very recently, a novel dense granule protein was identified by the Boothroyd laboratory that localizes to the PV and is processed approximately two-thirds along its sequence, revealing a C-terminal fragment that migrates at ~32 kDa and an N-terminal fragment that migrates at ~80 kDa (*Figure 6A*) (Franco et al., in press). Analysis of this protein, MYR1 (TGGT1_254470), revealed a TEXEL-like RRLSE sequence approximately 230 residues from the C-terminus, the approximate position where cleavage is predicted to occur (*Figure 6C*). We hypothesized that MYR1 is a substrate of ASP5 and to test this, we probed WT- and Δasp5-infected HFF lysates with antibodies derived to the N-terminal region of MYR1. In the Δasp5 mutants, MYR1 is no longer processed and instead migrates at ~105 kDa (*Figure 6A*). Further analysis of the ~80 kDa bands in the left panel using a 3–8% Tris-Acetate gel (right panel) revealed that this is a doublet, where the lower molecular weight species (*) is likely a cross-reactive protein often observed, even in knockout lines of this gene, by Franco et al. when using MYR1 antisera, but never when detecting this protein by C-terminal epitope tagging. This strongly suggests that ASP5 is required for cleavage of MYR1. Note that the predicted pI of MYR1 is ~5.0, which may contribute to the somewhat retarded mobility of the full-length and cleaved N-terminal species that have predicted masses of ~87 and 61 kDa, respectively).

We then tested whether MYR1 is a substrate of ASP5 by incubating immunoprecipitated $ASP5_{WT}$-$HA_3$ in combination with fluorogenic peptides containing RRLSE or mutations of this sequence. We found that $ASP5_{WT}$-$HA_3$ efficiently cleaved the MYR1 TEXEL peptide, similar to the GRA16 RRLAE control, whereas mutations of the conserved RRL residues abolished this activity (*Figure 6B*). This demonstrates that MYR1 contains a TEXEL sequence that can be processed by ASP5. It also suggests that the $P_{1'}$ residue, which naturally differs between GRA16 (alanine) and MYR1 (serine), is not as constrained as the $P_{1-3}$ positions, and confirms that the $P_2'$ residue is not essential for processing, since the MYR1 TEXEL $P_2'$ (E>A) mutant peptide was efficiently processed.

To test whether the TEXEL motif is necessary for cleavage in vivo, we generated lines that express MYR1 TEXEL mutants under control of the GRA1 promoter. While ectopic expression of the C-terminally tagged $MYR1_{WT}$-HA resulted in detection of the expected ~32 kDa species, mutation of either RRLSE>ARLSE or RRLSE>ARASA prevented cleavage, leaving only the unprocessed species migrating at ~105 kDa (*Figure 6D*). Thus, this sequence of MYR1 is necessary for processing in parasites. Taken together, these results demonstrate that ASP5 cleaves the TEXEL motif of MYR1, and that the TEXEL can function in a novel location near the C-terminus of the protein, in contrast to *Plasmodium* spp. where all known PEXEL sequences are located ~15–30 amino acids from the SP cleavage site (*Sargeant et al., 2006*).

## ASP5 influences host mitochondrial recruitment to the PVM

A striking feature of *Toxoplasma* infection is host mitochondrial association (HMA), whereby the parasite recruits host mitochondria to the PVM using the dense granule protein MAF1 that localizes to the PVM (*Pernas et al., 2014*). To examine whether ASP5 contributes to this phenotype, the ultrastructure of HFFs infected with WT and Δasp5 parasites was investigated by transmission electron microscopy (TEM). While the overall morphology of WT and Δasp5 tachyzoites appeared normal, there was a reduction in host mitochondria associated with the PVM of Δasp5-infected HFFs (*Figure 7A*). Quantification of HMA by TEM showed that the percentage of the PVM associated with host mitochondria was reduced by 4.3-fold in Δasp5 parasites (*Figure 7B*).

To confirm that the reduction in HMA observed by TEM in Δasp5-infected HFFs was due to the loss of ASP5, we used our $Δasp5_{CRISPR}$ parasites (*Figure 4A-ii*). These parasites were incubated for 4 hr on 60% confluent mouse embryonic fibroblasts (MEFs) engineered to express GFP fused to the mitochondrial targeting sequence (MTS) of DIABLO (MTS-GFP) (*Verhagen et al., 2000*). WT (parental RHΔhx) parasites efficiently associated with host mitochondria (MTS-GFP) and MAF1 was correspondingly observed at the PVM, as expected (*Figure 7C*, panel 1) (*Pernas et al., 2014*). In contrast, $Δasp5_{CRISPR}$ parasites exhibited reduced HMA, and MAF1 was incorrectly localized, appearing predominantly in punctate structures rather than at the PVM (*Figure 7C*, panels 2–4). In contrast, the $Δasp5_{CRISPR}$:$ASP5_{WT}$-$HA_3$ parasites exhibited correct trafficking of MAF1 and recruitment of host mitochondria to the PVM, which was validated in two independent complemented

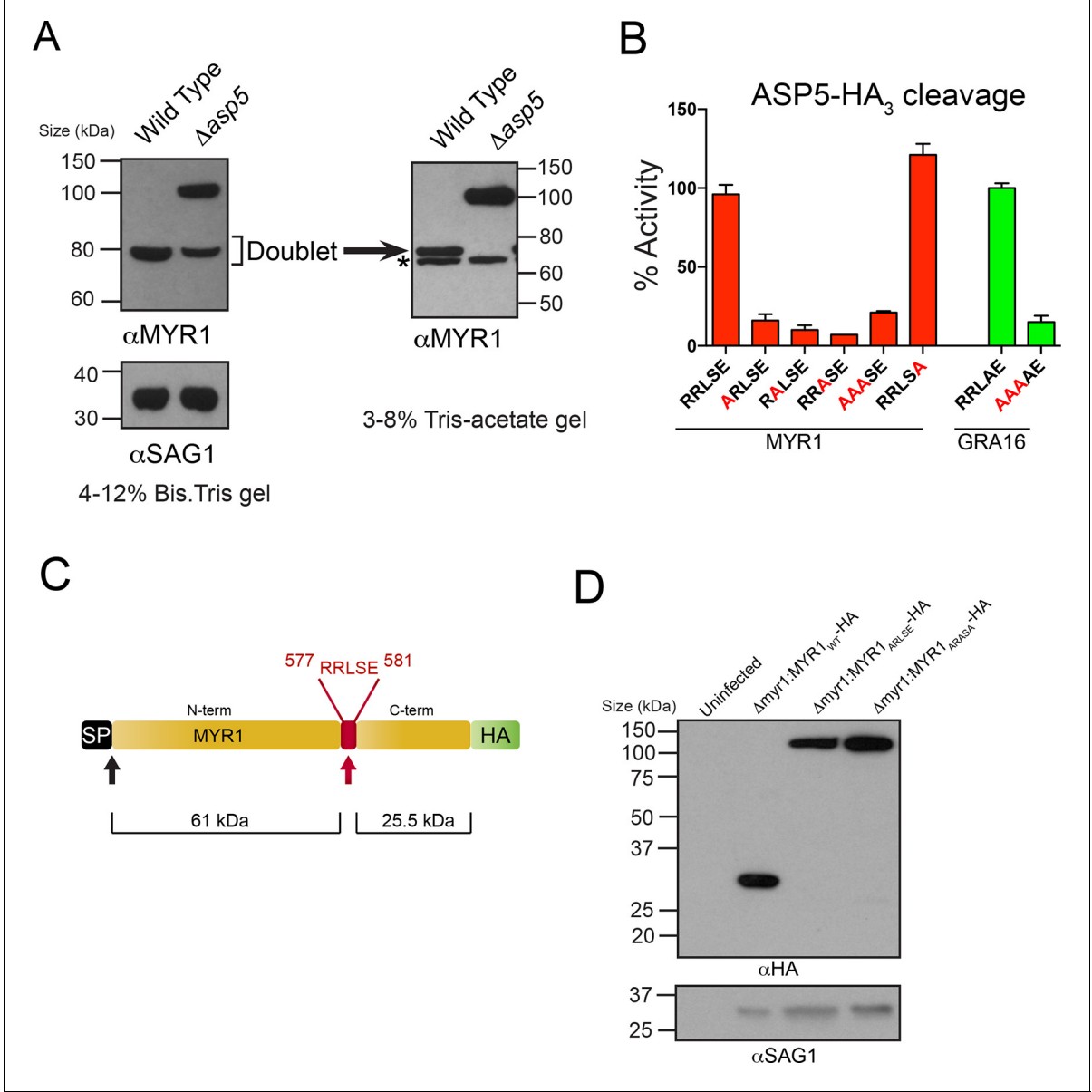

**Figure 6.** ASP5 processes the novel dense granule protein MYR1 near the C-terminus. (A) Antibodies to the N-terminus of MYR1 show that it is processed in wild-type parasites (migrating at ~80 kDa) and a loss of processing in Δasp5 parasites resulting in the appearance of a larger molecular weight species (migrating at ~105 kDa). αSAG1 serves as the parasite loading control. For increased resolution, the samples from the left panel were separated on a 3–8% Tris-acetate gel (right panel), which revealed that the ~80 kDa band migrates at ~70 kDa on this gel and comprises a doublet, with the upper band absent in Δasp5 parasites, confirming lack of cleavage. * = suspected cross-reactive species. As a consequence of increased running time, SAG1 migrated off the 3–8% gel and was not transferred. (B) ASP5$_{WT}$-HA$_3$ cleavage of DABCYL/EDANS peptides containing the TEXEL of MYR1 and associated mutations (red residues). Peptides containing RRLSE from MYR1 and RRLAE from GRA16 are cleaved, but peptides with point mutations in P$_1$, P$_2$ or P$_3$ are not cleaved. The serine at P$_1$' (compared to Ala in GRA16) does not interfere with cleavage by ASP5. (C) Schematic of MYR1 with an N-terminal SP, the RRLSE TEXEL at AA 557–581 and a C-terminal HA tag. (D) Immunoblot using αHA antibodies against Δmyr1:MYR1-HA parasites where Δmyr1:MYR1$_{WT}$-HA runs at ~32 kDa, whereas Δmyr1:MYR1$_{ARLSE}$-HA and Δmyr1:MYR1$_{ARASA}$-HA mutants run at ~105 kDa. αSAG1 serves as a loading control. ASP5, Aspartyl Protease 5; HA$_3$, triple-hemagglutinin; TEXEL, *Toxoplasma* export element.

clones that expressed ASP5 at levels close to the endogenous expression of the enzyme (*Figure 7C*, panels 5–6, and *Figure 4—figure supplement 1C*). While we observe changes in HMA in two independent Δasp5 mutants, observed by TEM and immunofluorescence, it should be noted that this phenotype appears to be somewhat variable when assayed in different labs.

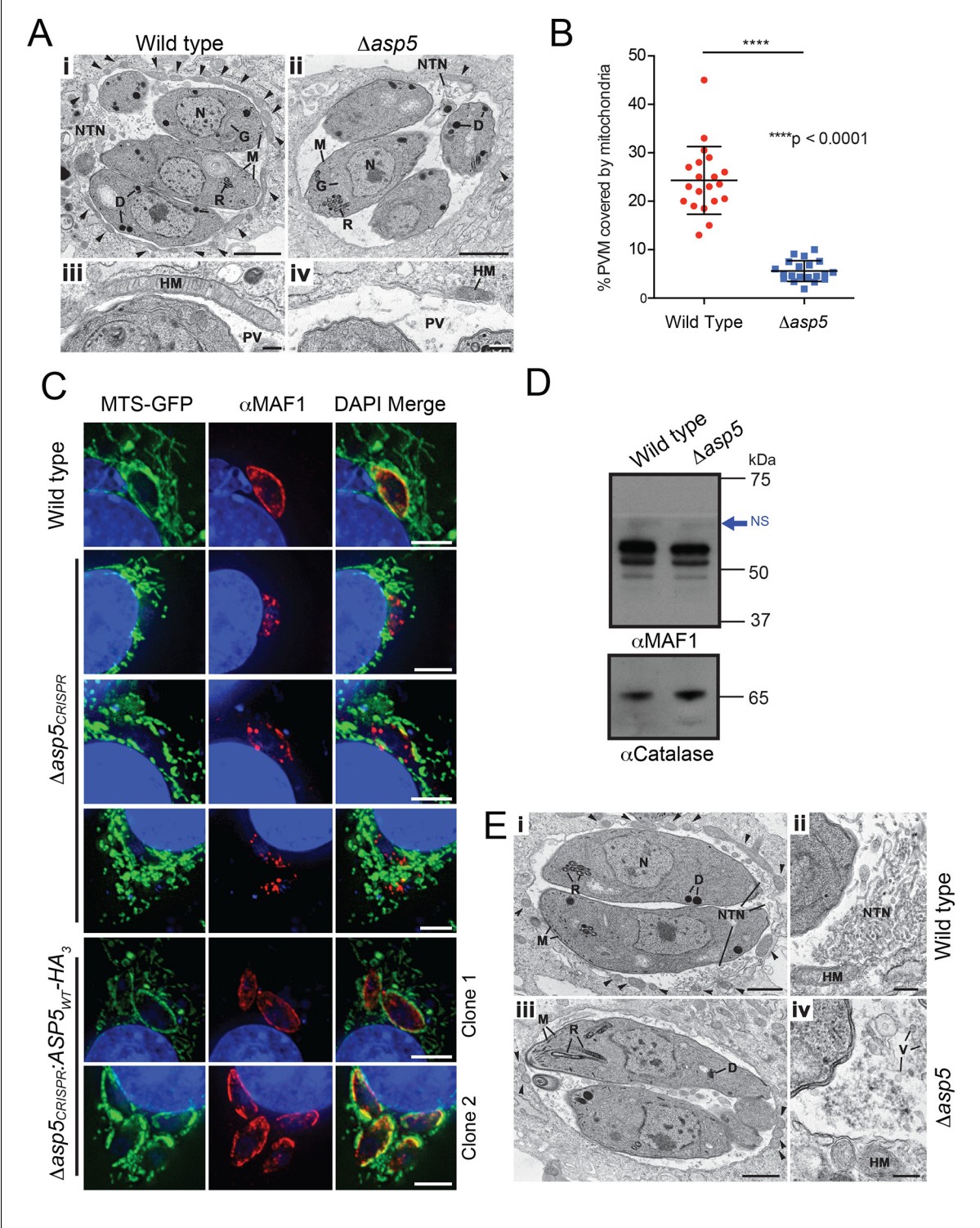

**Figure 7.** ASP5 influences efficient host mitochondrial recruitment and assembly of the NTN. (**A**) Electron micrographs of intracellular WT (Δ*ku80*) (i and iii) and Δ*asp5* (ii and iv) tachyzoites within HFFs. Bars represent 1 μm (i,ii) and 200 nm (iii, iv). (i, ii) Low-power image showing WT (i) and Δ*asp5* (ii) tachyzoites containing a nucleus (N), rhoptries (R), micronemes (M), dense granules (D) and a Golgi body (G) located within a PV. Note the large number of host cell mitochondria (arrowheads) associated with the PVM and the large NTN within the PV in wild-type parasites compared to Δ*asp5* parasites. (iii, iv) Details from the periphery of the PV showing a large host cell mitochondrion (HM) closely applied to the PVM in the wild type (iii) compared to the smaller mitochondrion (HM) associated with the Δ*asp5* PV (iv). (**B**) Quantitation of percentage of the PVM associated with host mitochondria, 5.59 ± 2.08% for Δ*asp5* parasites versus 24.3 ± 6.98% for wild-type parasites, mean ± standard error of the mean, *P* < 0.0001, n = 20

*Figure 7 continued on next page*

*Figure 7 continued*

vacuoles. (C) (i) Mouse embryonic fibroblasts expressing MTS-GFP infected for 4 hr with wild type (Δ*hx*), Δ*asp5*CRISPR (a non-GFP positive knock out) or two independent ASP5 complemented clones (Δ*asp5*CRISPR:ASP5WT-HA3). Localization of MAF1 at the PVM (top panel and bottom two panels) and mislocalized in intraparasitic puncta, potentially dense granules (panels 2 and 5), are shown in red. Mitochondria (MTS-GFP) are localized at the PVM in wild-type parasites (panel 1) and Δ*asp5*CRISPR:ASP5WT-HA3 clones 1 and 2 (panels 5–6) to a large extent, but less so in the Δ*asp5*CRISPR parasites (panels 2–4). (ii) Immunoblot using αHA antibodies against parasites expressing ASP5WT-HA3 and complemented mutants Δ*asp5*CRISPR:ASP5WT-HA3 clones 1 and 2 shows the parasites express similar levels of HA-tagged ASP5 (as in *Figure 2A*), αGAP45 serves as a loading control. (D) Western blot of MAF1 species in wild-type and Δ*ku80*Δ*asp5* parasites. Blue arrow shows non-specific labeling (NS), αCatalase serves as a loading control. (E) Electron micrographs of intracellular wild type (i and ii) and Δ*ku80*Δ*asp5* (iii and iv) tachyzoites. Bars represent 1 μm (i, iii) and 200 nm (ii, iv). (i, ii) Low-power image showing wild-type (i) and Δ*asp5* (iii) tachyzoites containing a nucleus (N), rhoptries (R), micronemes (M), and dense granules (D) located within the PV. The large number of host cell mitochondria (arrowheads) associated with the PVM and the large NTN within the PV in the wild type compared to the Δ*asp5* parasites is noteworthy. (ii) Detail of the PV of a WT parasite showing the intertwining tubules of the NTN. HM – host cell mitochondrion. (iv) Detail of the PV surrounding a Δ*asp5* parasite showing granular material and a few vesicles (V) but absence of the tubular network. HM – host cell mitochondrion. Scale bar is 5 μm. ASP5, Aspartyl Protease 5; GFP, green fluorescent protein; HFFs, human foreskin fibroblasts; MTS, mitochondrial targeting sequence; NTN, nanotubular network; PV, parasitophorous vacuole; PVM, parasitophorous vacuole membrane; WT, wild type.

MAF1 is not known to be proteolytically processed, beyond removal of its SP (*Pernas et al., 2014*). Nevertheless, to determine whether ASP5 affects the biosynthesis or processing of MAF1, we performed immunoblots with αMAF1 antibodies using WT and Δ*asp5* parasites. There were no differences in MAF1 expression or processing between the two lines by Western blot (*Figure 7D*), consistent with a lack of any TEXEL motif within MAF1. This result suggests that the function of MAF1 is not directly dependent on ASP5, but rather, ASP5 may act on as yet unidentified protein(s) that interact with MAF1 to facilitate efficient HMA.

## ASP5 is necessary for correct biogenesis of the NTN

Another characteristic of *Toxoplasma* infection is the formation of the NTN that likely aids nutrient acquisition across the PVM and within the PV through an increase in surface area. The ultrastructure of the NTN was examined in HFFs infected with WT and Δ*asp5* parasites by TEM (*Figure 7E*). Vacuoles containing WT parasites displayed extensive structures, extending from near the posterior of parasites to the PVM, typical of the NTN (*Figure 7E-i and ii*). In stark contrast, the NTN in Δ*asp5* vacuoles was vastly diminished and disorganized in all cells examined, suggesting that one or more components involved in the biogenesis of this network requires processing by ASP5 (*Figure 7E-iii and iv*).

## GRA24 requires ASP5 for export but does not possess a TEXEL

Following the identification of GRA24 as an exported effector protein that traffics to the host nucleus (*Braun et al., 2013*), we sought to determine whether its translocation into the host cell is also ASP5-dependent. WT and Δ*asp5* parasites were transfected with an ectopic copy of GRA24 fused to 3xMyc tags (GRA24-Myc3), which was integrated into the uracil phosphoribosyltransferase (URPT) locus. GRA24-Myc3 was expressed in parasites and exported to the host cell nucleus by WT parasites as previously described (*Braun et al., 2013*); however, export was completely blocked in Δ*asp5* parasites (*Figure 8A*, *Figure 8—figure supplement 1*). Complementation of Δ*asp5* parasites with ASP5WT-HA3 restored the export of GRA24-Myc3 (*Figure 8—figure supplement 1*). Despite the requirement of ASP5 for GRA24 export, assessment of processing by Western blot did not reveal any size difference in GRA24-Myc3 between WT and Δ*asp5* parasites (*Figure 8B*). While GRA24 lacks a canonical TEXEL sequence (RRLxx), it does contain the non-canonical TEXEL-like sequences RGYHG, RGGLQ and RSLGM, and so we assessed whether these might be cleaved by ASP5 using synthetic peptides; however, none were efficiently processed (*Figure 8C*). Collectively, this suggests that GRA24 is not a direct substrate of ASP5 but its export is dependent on this protease.

## ASP5 influences the expression of thousands of host genes during infection

Given our above findings, we wondered how important the ASP5-dependent pathway is to the transcriptional changes that *Toxoplasma* imparts on its host cell. Given that we determined there is little

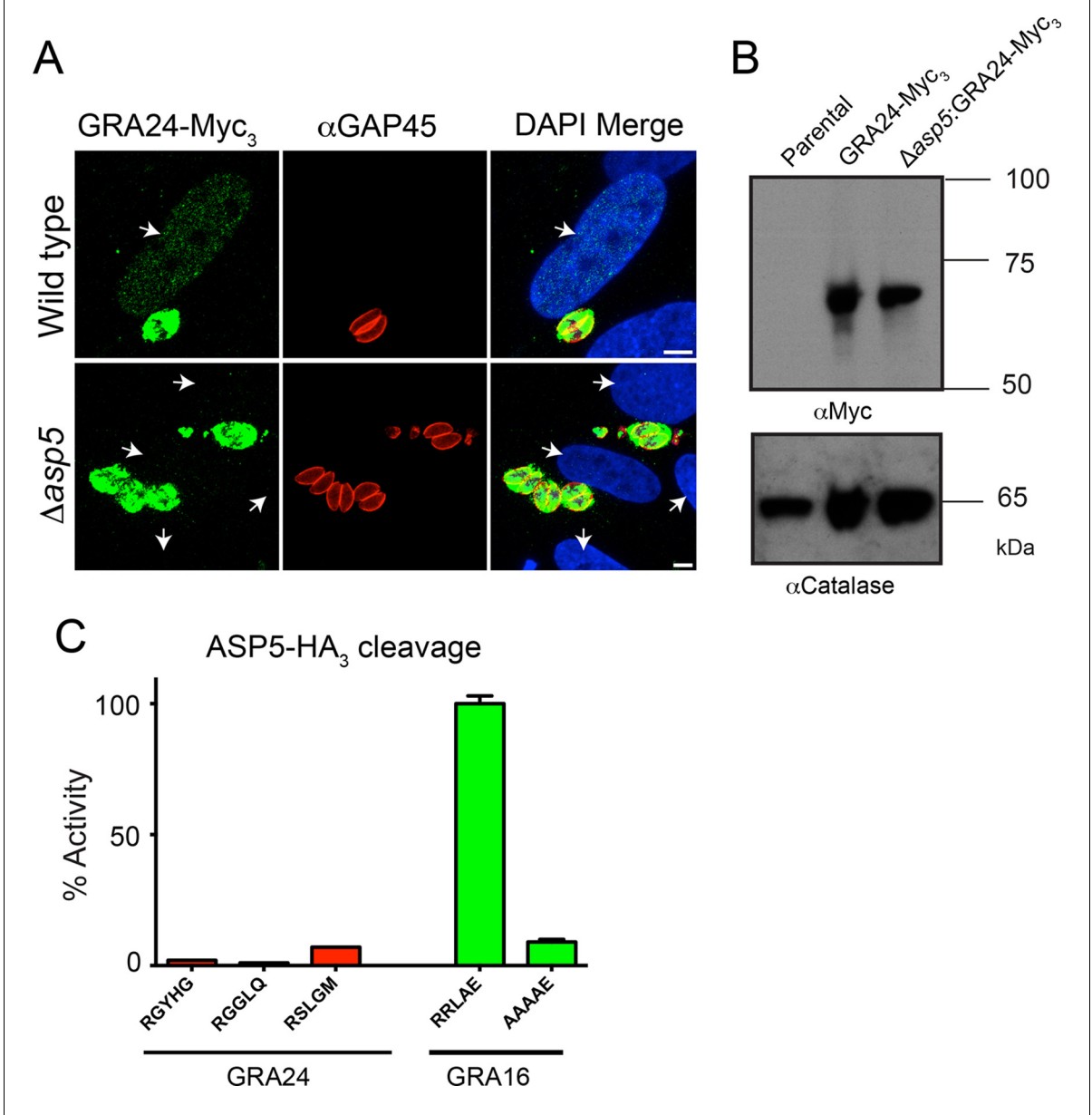

**Figure 8.** GRA24 requires ASP5 for export but not processing. (**A**) Localization of GRA24-Myc$_3$ in both WT (Δ*ku80*) and Δ*ku80*Δ*asp5* tachyzoites. GRA24 can be observed in the host nucleus and in the PV in WT:GRA24-Myc$_3$ parasites, whereas this export is lost in the Δ*asp5*:GRA24-Myc$_3$ parasites. Arrows signify the position of host nuclei (DAPI). (**B**) The size of GRA24-Myc$_3$ appears unchanged in the absence of ASP5. αCatalase serves as a loading control. (**C**) ASP5 cannot cleave peptides containing non-canonical TEXEL-like motifs found within GRA24, compared with cleavage of the GRA16 RRLAE peptide and AAAAE controls. Scale bar is 5 μm. ASP5, Aspartyl Protease 5; DAPI, 4′,6-diamidino-2-phenylindole; PV, parasitophorous vacuole; TEXEL, *Toxoplasma* export element; WT, wild type.

The following figure supplement is available for figure 8:

**Figure supplement 1.** Complementation of Δasp5 parasites restores export of GRA24.

to no change in replication rates between WT and Δ*asp5* parasites (**Figure 4B**), we harvested all samples 20 hr after infection and used RNA sequencing (RNA-seq) to profile gene expression in HFFs that were either uninfected (UI), infected with WT (RHΔ*ku80*) parasites, or infected with Δ*asp5* parasites. To make sure that all the changes that we observed were due to loss of ASP5 and not differences in tachyzoites numbers, we first compared the proportion of reads (rpkm) from parasite

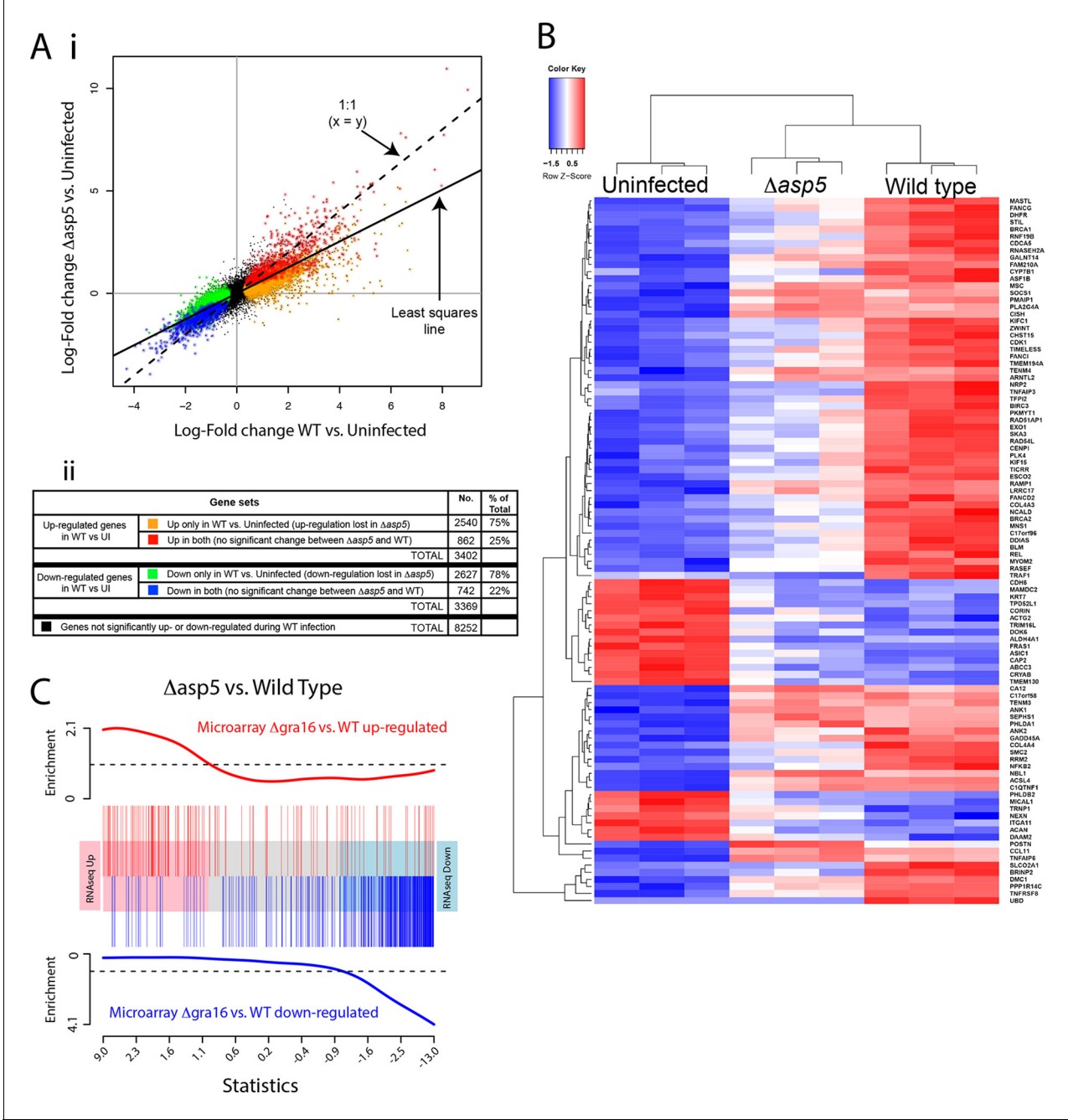

**Figure 9.** ASP5 plays a major role in changing the host cell transcriptional response induced by *Toxoplasma* infection. (**A**) (i) Scatterplot of expression fold changes. The Y-axis shows log2-fold changes in HFFs infected with Δ*asp5* parasites versus uninfected HFFs (UI), while the X-axis shows log2-fold changes in HFFS infected with WT parasites (WT) vs. UI. The dashed line shows x=y. The solid line shows the least squares regression line through the origin. The regression has slope 0.6, showing that log fold changes for the Δ*asp5* parasites are only 60% of those for the wild-type parasites. Differentially expressed genes are color coded in the plot according to whether they change in both the WT and Δ*asp5* infections or only in the WT (false discovery rate < 0.05). Non-differentially expressed genes are shown in black. (ii) Numbers of genes corresponding to highlighted groups in the scatterplot. (**B**) Heat map of expression values for the 100 most differentially expressed genes for WT-infected HFFs versus uninfected. Z-scores are log2 counts per million, scaled to have mean 0 and standard deviation 1 for each gene. The plot shows that expression after Δ*asp5* infection tends to be intermediate between that of uninfected and WT-infected HFFs. (**C**) Barcode enrichment plot showing enrichment of Δ*gra16* regulated genes in the Δ*asp5* parasite infection expression changes. Genes are ordered from left to right in the plot from most up to most down during Δ*asp5* parasite infection. Specifically, genes are ranked from largest to smallest t-statistic for the Δ*asp5* versus WT comparison (X-axis). Genes up-regulated by Δ*gra16* versus WT in an independent experiment (*Bougdour et al., 2013*) are marked with vertical red bars. Similarly, genes down-regulated by Δ*gra16* in the independent experiment are marked with vertical blue bars. The worms show relative enrichment (Y-axes). The plot shows that Δ*asp5* up-regulated

*Figure 9 continued on next page*

Figure 9 continued

genes are strongly enriched for Δgra16 up-regulated genes (red) and Δasp5 down-regulated genes are strongly enriched for Δgra16 down-regulated genes (blue). HFFs, human foreskin fibroblasts; WT, wild type.

versus host cell origin as a readout of relative parasite numbers per sample. We saw equal amounts of reads mapping to human genes between all samples ($24$–$27 \times 10^6$ reads), while infection with WT saw parasite RNA proportions of 27% (replicate 1), 24% (replicate 2) and 23% (replicate 3). Infection with our ASP5 deficient line saw parasite RNA to be 18% (replicate 1), 35% (replicate 2) and 36% (replicate 3) of the total reads, therefore suggesting that, in 2 out of the 3 samples, we have slightly more ASP5-deficient parasites per sample. Therefore, any loss of gene expression in ASP5-deficient cells must be due to loss of this protease and not lower amounts of overall parasites per sample. The expression changes due to infection by the Δasp5 parasites were generally smaller than those for the WT parasites. The log-fold change during infection with Δasp5 parasites was, on average, only 60% of the log-fold change during infection with WT parasites (*Figure 9A-i*). This suggests that most genes responding to parasite infection do so, at least partly, due to an ASP5-dependent pathway. At a false discovery rate of 5%, 3402 genes were significantly up-regulated and 3369 genes were significantly down-regulated in response to the WT infection, whereas only 1033 genes were significantly up- and 817 were significantly down-regulated in response to Δasp5 parasites. Of the 3402 genes up-regulated during WT infection, only 862 (25% ) remained significantly up-regulated upon deletion of *ASP5* (*Figure 9A-ii*). Of the 3269 genes down-regulated during WT infection, only 742 (22%) remained significantly down-regulated upon deletion of *ASP5* (*Figure 9A-ii*). This identifies genes (color-coded red and blue in *Figure 9A*) that are potentially unaffected by ASP5-dependent pathways.

To further analyze the role of ASP5-dependent export pathways on transcriptional changes, we extracted the 100 most differentially expressed genes upon WT infection (compared with UI) and generated a heat map to reveal the contribution of ASP5 to the expression of these genes by comparing with the dataset derived using Δasp5 parasites (*Figure 9B*). As expected, the three biological replicates cluster together well for each condition. The plot shows that expression in HFFS infected with Δasp5 parasites tends to be intermediate between uninfected cells and the WT infection (*Figure 9B*). Overall, this work suggests that ASP5-dependent pathways contribute significantly to the amount and magnitude of expression of host cell genes during tachyzoite infection.

To assess the role the ASP5-dependent pathways in modifying the host cell, we performed Gene Ontology (GO) analysis on gene subsets as listed above (*Table 1*; color-coded as in *Figure 9A-ii*). We observed that ASP5 controlled the up-regulation of gene sets implicated in cell cycle, nucleic acid metabolism and binding, nucleopore association and chromatin binding. Furthermore, ASP5 played a key role in the down-regulation of genes implicated in autophagy, peroxisome fission, vacuole organization, protein trafficking (i.e. syntaxin binding) and intracellular signaling processes (*Table 1*). This outlines that ASP5-dependent pathways play an important role in controlling specific cellular processes that may facilitate parasite persistence within the cell.

As ASP5 affects the processing and translocation of GRA16, we hypothesized that transcriptional changes induced by the loss of ASP5 would encompass the changes caused by this single effector. We obtained a list of genes that are differentially expressed in HFFs infected with Δgra16 versus WT parasites from a previously published study (*Bougdour et al., 2013*). We found that the transcriptional profile of Δgra16 parasite infection is strongly correlated with the transcriptional profile that we observed in Δasp5 parasite infection. Genes up-regulated in the Δasp5 parasite infection were strongly enriched for up-regulated Δgra16 genes, and similarly, the down-regulated Δasp5 were strongly enriched for Δgra16 genes (ROAST P-value =$5 \times 10^{-5}$). *Figure 9C* shows the enrichment as a barcode plot. This shows that transcriptional changes induced by GRA16 mostly represent a subset of all genes influenced by ASP5.

## ASP5 is an important virulence factor

Given the multiple effects that ASP5 plays on the cellular changes and transcriptional output of the infected host cell, we sought to determine whether this Golgi-resident protein, and the export

**Table 1.** Gene ontology analysis of expression changes following infection with WT and Δasp5 parasites. The four columns of the table correspond to the color-coded gene groups shown in **Figure 8A**. For each group of genes, the table gives the top 10 BP and MF represented in the differentially expressed genes.

| | | | | | | | | | | | | |
|---|---|---|---|---|---|---|---|---|---|---|---|---|
| | **Host gene expression significantly affected by loss of ASP5** | | | | | | | | | | | |
| | **DE genes that are up-regulated in wild type versus uninfected only** | | | | | | **DE genes that a down-regulated in wild type versus uninfected only** | | | | | |
| Biological process (BP) | GO ID | Term | Ont | N | DE | P.DE | GO ID | Term | Ont | N | DE | P.DE |
| | GO:0000278 | mitotic cell cycle | BP | 884 | 284 | 2.97E-26 | GO:0006914 | autophagy | BP | 295 | 79 | 1.40E-05 |
| | GO:0090304 | nucleic acid metabolic process | BP | 3990 | 918 | 8.82E-23 | GO:0010927 | cellular component assembly involved in morphogenesis | BP | 192 | 56 | 1.85E-05 |
| | GO:0022402 | cell cycle process | BP | 1096 | 323 | 1.08E-22 | GO:0000045 | autophagic vacuole assembly | BP | 60 | 24 | 2.03E-05 |
| | GO:0007049 | cell cycle | BP | 1446 | 401 | 1.15E-22 | GO:0016559 | peroxisome fission | BP | 10 | 8 | 2.26E-05 |
| | GO:1903047 | mitotic cell cycle process | BP | 772 | 243 | 3.91E-21 | GO:0042594 | response to starvation | BP | 173 | 51 | 3.13E-05 |
| | GO:0006139 | nucleobase-containing compound metabolic process | BP | 4441 | 995 | 5.21E-21 | GO:0044782 | cilium organization | BP | 145 | 44 | 5.00E-05 |
| | GO:1901360 | organic cyclic compound metabolic process | BP | 4710 | 1040 | 5.98E-20 | GO:0007033 | vacuole organization | BP | 110 | 35 | 1.00E-04 |
| | GO:0022613 | ribonucleoprotein complex biogenesis | BP | 310 | 123 | 8.72E-20 | GO:0051146 | striated muscle cell differentiation | BP | 169 | 48 | 1.45E-04 |
| | GO:0006725 | cellular aromatic compound metabolic process | BP | 4559 | 1009 | 1.84E-19 | GO:0030031 | cell projection assembly | BP | 269 | 69 | 1.97E-04 |
| | GO:0006396 | RNA processing | BP | 532 | 180 | 2.08E-19 | GO:1903008 | organelle disassembly | BP | 162 | 46 | 1.98E-04 |
| Molecular function (MF) | GO ID | Term | Ont | N | DE | P.DE | GO ID | Term | Ont | N | DE | P.DE |
| | GO:0044822 | poly(A) RNA binding | MF | 1114 | 380 | 4.06E-42 | GO:0017049 | GTP-Rho binding | MF | 14 | 9 | 0.000104 |
| | GO:0003723 | RNA binding | MF | 1445 | 452 | 2.22E-39 | GO:0033743 | peptide-methionine (R)-S-oxide reductase activity | MF | 4 | 4 | 0.000837 |
| | GO:0003676 | nucleic acid binding | MF | 3243 | 795 | 7.02E-28 | GO:0004030 | aldehyde dehydrogenase [NAD (P)+] activity | MF | 5 | 4 | 0.003616 |
| | GO:1901363 | heterocyclic compound binding | MF | 4739 | 1043 | 1.67E-19 | GO:0030553 | cGMP binding | MF | 5 | 4 | 0.003616 |
| | GO:0097159 | organic cyclic compound binding | MF | 4780 | 1048 | 5.08E-19 | GO:0004499 | N,N-dimethylaniline monooxygenase activity | MF | 5 | 4 | 0.003616 |
| | GO:0003682 | chromatin binding | MF | 383 | 110 | 1.05E-07 | GO:0019905 | syntaxin binding | MF | 65 | 20 | 0.004494 |
| | GO:0043566 | structure-specific DNA binding | MF | 217 | 67 | 2.22E-06 | GO:0031697 | beta-1 adrenergic receptor binding | MF | 3 | 3 | 0.004923 |
| | GO:0017056 | structural constituent of nuclear pore | MF | 9 | 8 | 8.09E-06 | GO:0045159 | myosin II binding | MF | 3 | 3 | 0.004923 |
| | GO:0005488 | binding | MF | 10573 | 1974 | 8.11E-06 | GO:0047555 | 3',5'-cyclic-GMP phosphodiesterase activity | MF | 3 | 3 | 0.004923 |
| | GO:0008094 | DNA-dependent ATPase activity | MF | 76 | 30 | 8.31E-06 | GO:0031210 | phosphatidylcholine binding | MF | 8 | 5 | 0.00505 |

*Table 1 continued on next page*

Coffey *et al.* eLife 2015;4:e10809. DOI: 10.7554/eLife.10809

*Table 1 continued*

| | Host gene expression not affected by loss of ASP5 | | | | | | | | | | |
|---|---|---|---|---|---|---|---|---|---|---|---|
| | DE genes that are up-regulated in both wild type versus uninfected and *?asp5* versus uninfected | | | | | | DE genes that are down-regulated in both wild type versus uninfected and *?asp5* versus uninfected | | | | |
| Biological process (BP) | GO ID | Term | Ont | N | DE | P.DE | GO ID | Term | Ont | N | DE | P.DE |
| | GO:0044699 | single-organism process | BP | 9400 | 689 | 1.02E-20 | GO:0003008 | system process | BP | 888 | 87 | 5.74E-10 |
| | GO:0044763 | single-organism cellular process | BP | 8542 | 642 | 5.49E-20 | GO:0032501 | multicellular organismal process | BP | 4364 | 286 | 6.37E-09 |
| | GO:0050896 | response to stimulus | BP | 5333 | 443 | 4.11E-17 | GO:0044707 | single-multicellular organism process | BP | 4228 | 277 | 1.49E-08 |
| | GO:0032501 | multicellular organismal process | BP | 4364 | 365 | 2.73E-13 | GO:0006928 | movement of cell or subcellular component | BP | 1295 | 104 | 4.60E-07 |
| | GO:0044707 | single-multicellular organism process | BP | 4228 | 354 | 8.10E-13 | GO:0045216 | cell-cell junction organization | BP | 176 | 26 | 5.96E-07 |
| | GO:0006950 | response to stress | BP | 2675 | 246 | 1.84E-12 | GO:0034330 | cell junction organization | BP | 205 | 28 | 1.13E-06 |
| | GO:0051716 | cellular response to stimulus | BP | 4476 | 367 | 4.63E-12 | GO:0048731 | system development | BP | 2914 | 195 | 1.93E-06 |
| | GO:0032502 | developmental process | BP | 3950 | 331 | 8.70E-12 | GO:0048513 | organ development | BP | 2054 | 143 | 1.01E-05 |
| | GO:0065007 | biological regulation | BP | 7817 | 570 | 2.04E-11 | GO:0034329 | cell junction assembly | BP | 182 | 24 | 1.20E-05 |
| | GO:0042221 | response to chemical | BP | 2547 | 232 | 3.00E-11 | GO:0044767 | single-organism developmental process | BP | 3888 | 243 | 1.25E-05 |
| Molecular function (MF) | GO ID | Term | Ont | N | DE | P.DE | GO ID | Term | Ont | N | DE | P.DE |
| | GO:0005125 | cytokine activity | MF | 103 | 21 | 9.63E-07 | GO:0008092 | cytoskeletal protein binding | MF | 635 | 61 | 5.31E-07 |
| | GO:0000982 | RNA polymerase II core promoter proxi-mal region sequence-specific DNA binding transcription factor activity | MF | 201 | 31 | 1.93E-06 | GO:0003779 | actin binding | MF | 299 | 36 | 7.99E-07 |
| | GO:0008009 | chemokine activity | MF | 22 | 9 | 2.92E-06 | GO:0022836 | gated channel activity | MF | 164 | 24 | 1.91E-06 |
| | GO:0005515 | protein binding | MF | 8144 | 560 | 6.08E-06 | GO:0004872 | receptor activity | MF | 592 | 53 | 2.29E-05 |
| | GO:0043565 | sequence-specific DNA binding | MF | 567 | 62 | 6.57E-06 | GO:0005216 | ion channel activity | MF | 202 | 25 | 2.40E-05 |
| | GO:0000981 | sequence-specific DNA binding RNA polymerase II tran-scription factor activity | MF | 357 | 44 | 7.96E-06 | GO:0022838 | substrate-specific channel activity | MF | 204 | 25 | 2.84E-05 |
| | GO:0004857 | enzyme inhibitor activity | MF | 226 | 32 | 8.47E-06 | GO:0015267 | channel activity | MF | 217 | 25 | 7.97E-05 |
| | GO:0044212 | transcription regulatory region DNA binding | MF | 457 | 52 | 1.24E-05 | GO:0022803 | passive transmembrane transporter activity | MF | 217 | 25 | 7.97E-05 |
| | GO:0000975 | regulatory region DNA binding | MF | 459 | 52 | 1.40E-05 | GO:0038023 | signaling receptor activity | MF | 440 | 41 | 8.14E-05 |
| | GO:0001067 | regulatory region nucleic acid binding | MF | 459 | 52 | 1.40E-05 | GO:0005230 | extracellular ligand-gated ion channel activity | MF | 25 | 7 | 1.60E-04 |

ASP5, Aspartyl Protease 5; BP, biological processes; DE = number of those genes that are differentially expressed genes; GO, Gene Ontology; MF, molecular functions; N = number of expressed genes annotated by the GO term; P = p-value; WT, wild type.

pathway that it controls, are important virulence mechanisms in *Toxoplasma*. To determine this, we injected groups of 6 C57BL/6 mice with either phosphate buffered saline (PBS) or 100 WT, $\Delta asp5_{CRISPR}$ or $\Delta asp5_{CRISPR}$:ASP5$_{WT}$-HA$_3$ parasites (*Figure 10A*), which equated to 15 ± 3 live tachyzoites (as determined by in vitro plaque assay). All mice were tested for sero-conversion to confirm the administration of parasites (*Figure 10—figure supplement 1A*) and any that did not elicit a response were discounted from the study (this equated to two mice for each group). Over a 20-day period, we found that all mice infected with either WT or $\Delta asp5_{CRISPR}$:ASP5$_{WT}$-HA$_3$ tachyzoites succumbed to infection by day 8 and dropped weight accordingly (*Figure 10A*). Strikingly, all mice infected with $\Delta asp5_{CRISPR}$ parasites were still alive at day 20, they maintained body weight and appeared healthy, despite sero-converting by day 14 (*Figure 10—figure supplement 1*). We also confirmed prior infection by performing a re-challenge experiment, where mice were injected with 200 wild-type parasites, equating to ~50 live parasites (as determined by plaque assay). While 'naïve' PBS (vehicle)-injected mice succumbed to infection by day 10, all those previously injected with $\Delta asp5_{CRISPR}$ all survived and maintained normal body weight (*Figure 10B*). To determine whether the attenuation of $\Delta asp5_{CRISPR}$ parasites in mice was dependent on the infectious dose, an additional cohort of C57/BL6 mice was injected with ~50 live parasites of each strain, as determined by in vitro plaque assay (*Figure 10C*). All mice infected with wild type or $\Delta asp5_{CRISPR}$:ASP5WT-HA$_3$ tachyzoites succumbed to infection by day 10, whereas those injected with $\Delta asp5_{CRISPR}$ parasites exhibited a delay to death, including one mouse that survived the experiment and was seropositive for anti-*Toxoplasma* antibodies when tested at day 14 (*Figure 10—figure supplement 1B*).

Thus, ASP5-deficient tachyzoites exhibit attenuation even in the hypervirulent RH strain at a dose of 50 parasites, while injection with 15 parasites resulted in parasite clearance and provided protective immunity following re-challenge with a lethal dose of wild type tachyzoites. This work strongly suggests that the ASP5-dependent export pathway is necessary for virulence of *Toxoplasma* in a mouse model.

## Discussion

*Toxoplasma gondii* has the remarkable capacity to persist within almost any nucleated host cell in a vast array of organisms. Central to this is the ability to manipulate host cellular pathways using exported effector proteins to circumvent the host response to allow for intracellular parasite growth and survival. Previous work has demonstrated that *Toxoplasma* effectors can be delivered to the host cell by secretion from the rhoptry organelles (*Koshy et al., 2010*; *Saeij et al., 2007*; *Boothroyd and Dubremetz, 2008*). These effectors largely consist of a family of kinases and appear to be exclusively delivered to the host cell during the short time frame of host cell invasion, potentially limiting their efficacy later during the infection process (*Saeij et al., 2007*; *Peixoto et al., 2010*). While injection of these polymorphic kinases explained some strain variances in virulence, it did not clarify how *Toxoplasma* induces changes that are more general across isolates. More recently, several new host cell effectors have been identified that appear to be delivered into the host cell via the dense granules—organelles constitutively secreted after invasion during intracellular replication (*Bougdour et al., 2013*; *Braun et al., 2013*; *Pernas et al., 2014*; *Rosowski et al., 2011*). This strongly suggested that *Toxoplasma* utilizes two export pathways; the rhoptry secretion pathway, which operates early during infection, and the dense granule export pathway, which we characterize here and show is dependent on ASP5 activity.

Upon the recent identification of GRA16, a dense granule effector that translocates into the host cell nucleus and affects p53 turnover, we noticed a PEXEL-like sequence at the approximate location that the PEXEL motif is located in *P. falciparum* proteins (*Bougdour et al., 2013*; *Hiller, 2004*; *Marti, 2004*). Our work described here shows that this TEXEL motif is involved in the export of GRA16 into the host cell and is processed by the Golgi-resident ASP5, consistent with a recent publication characterizing ASP5 (*Curt-Varesano et al., 2015*). While we identified this system based on its similarity to the *Plasmodium* export pathway, and indeed it has now come to light that several other Apicomplexan species utilize this 'PEXEL-like motif' for protein export (*Pellé et al., 2015*), our work has uncovered important differences between the *Toxoplasma* and *Plasmodium* systems and therefore sheds new light on protein export by Apicomplexan parasites.

In this study, the consensus substrate sequence for ASP5 was determined to be RRLxx, demonstrating that this enzyme has different substrate specificity to PMV, which requires RxLxE/Q/D for

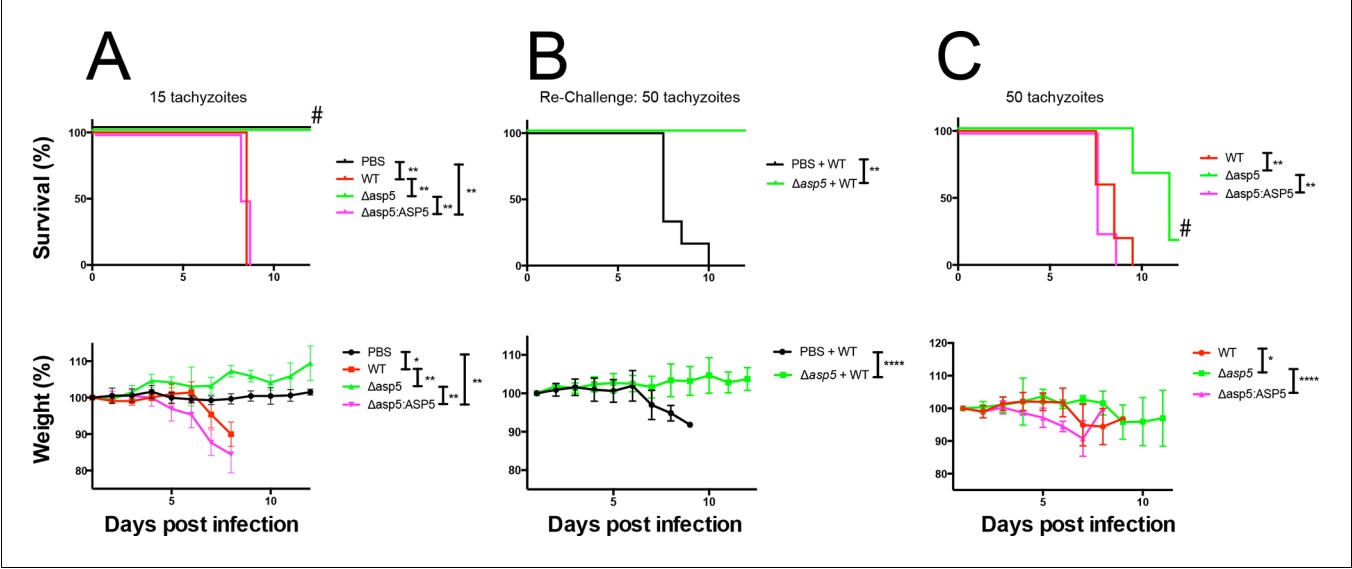

**Figure 10.** ASP5 is an important virulence factor. (A) Four groups of six C57BL/6 mice were intraperitoneally injected with a live dose of 15 ± 3 tachyzoites or PBS alone and survival measured over a 20-day period. Mice infected with wild type (RHΔ*hx*) and Δasp5$_{CRISPR}$:ASP5$_{WT}$-HA$_3$ all succumbed to infection within 8 days, whereas all PBS-injected mice and those infected with Δ*asp5*$_{CRISPR}$ parasites survived the 20 day experiment. At day 14 (#), all mice were bled and tested for antibodies against tachyzoites. Animals were weighed daily throughout the course of the experiment (lower panel) and bodyweights were compared for statistical analysis while all animals were alive. Mice injected with PBS alone maintained a stable body weight, while those infected with wild type and Δasp5$_{CRISPR}$:ASP5$_{WT}$-HA$_3$ parasites lost weight beginning at day 6 and day 4, respectively, with significant weight loss evident in comparison to those injected with *Dasp5*$_{CRISPR}$ parasites by day 7. (B) At 24 days post infection, surviving mice (from A) that were injected with PBS or Δasp5$_{CRISPR}$ tachyzoites were re-challenged with 50 live RHΔ*hx* parasites. The naïve PBS-injected mice all succumbed to infection by day 10, whereas those that had been injected with Δasp5$_{CRISPR}$ parasites were protected from death. Bodyweight was also monitored daily (lower panel) where mice previously injected with Δasp5$_{CRISPR}$ parasites maintained a stable bodyweight, while the naïve PBS mice began losing weight on approximately day 6. (C) A separate cohort of C57/BL6 mice was also injected with 50 live parasites to assess the effect of parasite number on virulence during infection. All mice infected with WT or and Δasp5$_{CRISPR}$:ASP5$_{WT}$-HA$_3$ parasites again succumbed to infection by days 8-10, whereas there was a delay in death for the and Δasp5$_{CRISPR}$-infected mice. One of these mice survived the 15-day experiment and was seropositive for antibodies against *Toxoplasma* (serum collected at day 14, #). Bodyweights were measured daily (lower panel). Log-rank (Mantel-Cox) testing was used to derive statistical significance for survival curves while two-way analysis of variance testing was used for bodyweight data. Values are mean ± SD. * P < 0.05, ** P < 0.005, **** P <0.0001. ASP5, Aspartyl Protease 5; WT, Wild Type.

The following figure supplement is available for figure 10:

**Figure supplement 1.** Seroconversion of Δ*asp5 parasites.*

activity (*Boddey et al., 2009*; *2010*; *Hodder et al., 2015*; *Sleebs et al., 2014b*). A small molecule compound that mimics RRL and contains the non-cleavable amino acid statine (WEHI-586), thus a likely competitive inhibitor, inhibited ASP5 activity, whereas WEHI-916, a potent PEXEL-mimetic inhibitor of PMV that contains valine at P$_2$ (RVL$_{statine}$) (*Sleebs et al., 2014b*) was a very poor ASP5 inhibitor (IC$_{50}$ >20 µM; data not shown). Our structural model of ASP5 in complex with the TEXEL of GRA16 provides a clear explanation for the requirement of RRL within the substrate. Interestingly, we observed that mutation of the P$_{2'}$ residue increased ASP5 activity. This likely reflects a reduced entropic penalty associated with forming the salt bridge between the otherwise flexible side-chains. The ASP5 model also revealed differences between it and PMV, including the absence of a helix-turn-helix motif in PMV that is thought to participate in protein–protein interactions in the ER, consistent with ASP5's location in the Golgi.

The PEXEL in *P. falciparum* is usually found in close proximity to the N-terminal signal peptide; however, we have shown here that this positional constraint does not occur in the *Toxoplasma* TEXEL. We demonstrate that MYR1, a novel protein essential secreted into the PV, has an ASP5-cleaved TEXEL motif approximately 558 amino acids from the predicated signal peptidase cleavage

site. Plasmepsin V of *Plasmodium spp.* is an ER-resident enzyme that cleaves the PEXEL co-translationally (*Sleebs et al., 2014b*), potentially residing in complex with signal peptidase, thus immediately licensing proteins for export upon ER entry. This cannot occur for ASP5, as it is located in the Golgi, which would require TEXEL-cargo proteins to be trafficked via vesicles to this compartment prior to its activity. It is interesting to note that this localization for ASP5 is not unique as other aspartyl proteases involved in protein trafficking, such as furin and the beta-site amyloid precursor protein cleaving enzyme (BACE), are located in the Golgi (*Thomas, 2002*; *Evin et al., 2010*). It is therefore possible that the positional constraint of the *Plasmodium* PEXEL within cargo proteins is the consequence of PMV's ER localization, whereas ASP5's location in the Golgi may permit cleavage of a TEXEL motif at any position within a substrate protein. It is possible, however, that PMV may also cleave PEXEL motifs found anywhere within *Plasmodium* proteins containing a signal peptide, as this is yet to be investigated. An important difference between the *Plasmodium* and *Toxoplasma* export pathways is that, while PEXEL cleavage appears to be solely involved in export, the apparent lack of export of cleaved MYR1 suggests that TEXEL processing may also be necessary for correct localization and/or function in the PV. We also found that Δ*asp5* parasites displayed a profound defect in the biogenesis of the NTN within the PV, which is known to require several PV proteins of dense granule origin. The NTN resides between replicating parasites and the PVM, where it potentially functions in the exchange of solutes by increasing surface area (*Mercier, 2002*). It is presently unknown whether the genesis of the NTN requires proteins that are exported into the host cell but it is interesting to note that GRA14, a protein known to be involved in NTN formation, has a putative RRLxx motif (*Rome et al., 2008*).

We also show that trafficking of proteins that lack discernable TEXEL motifs are affected by deletion of *ASP5*. We show that mitochondrial recruitment, which depends on the dense granule protein MAF1, is reduced in parasites lacking *ASP5*. We also show that ASP5 is essential for the export of GRA24, an effector that promotes sustained MAPK signaling within the host cell (*Braun et al., 2013*). Some *Plasmodium* exported proteins do not have a PEXEL motif and instead rely on a transmembrane domain and other unknown factors for translocation through the PTEX translocon. These PEXEL-negative exported proteins (PNEPs) include the major virulence protein PfEMP1 and several proteins required for the transport of this large molecule to its final destination on the erythrocyte surface for cytoadherence (*Maier et al., 2008*; *Rug et al., 2014*; *Sleebs et al., 2014b*). The export of PfEMP1 is also dependent on several PEXEL-containing proteins (*Maier et al., 2008*, *Rug et al., 2014*) as well as PMV (*Sleebs et al., 2014b*). Thus, our current hypothesis is that trafficking of GRA24, MAF1 and potentially other TEXEL-negative proteins rely on one or more TEXEL-containing proteins.

To understand the importance of ASP5-dependent export pathways on inducing transcriptional changes within the host cell we performed RNA-seq experiments and analyzed differences in up- and down-regulated genes induced by infection with wild type or Δ*asp5* parasites. We found that loss of ASP5 results in a global reduction in the magnitude of host cell transcriptional changes in response to parasite infection. By interrogating the biological processes and molecular functions of genes that are influenced by ASP5, it is evident that this protease plays an important role in influencing the expression of genes involved in cell cycle, nucleic acid metabolism, autophagy, peroxisome fusion, vacuole organization, cell differentiation, signaling processes and proteins that bind DNA and chromatin (*Table 1*). While GRA16 has been implicated in cell cycle progression and GRA24 influences transcription factor expression, there are as yet no known effectors that influence the other characterized biological processes. It is clear that understanding how *Toxoplasma* influences these in the infected cell is an important step. Furthermore, profiling the transcriptional changes that occur in other cell types that *Toxoplasma* is known to infect, such as macrophages, dendritic cells, myocytes and neurons, and the ability to determine the influence of ASP5-dependent pathways on these changes is now imminently achievable.

It is noteworthy that ASP5 could be deleted from the genome of *Toxoplasma*, demonstrating that this enzyme in not essential, unlike PMV, which cannot be genetically deleted using conventional approaches (*Boddey et al., 2010*; *Klemba and Goldberg, 2005*; *Russo et al., 2010*; *Sleebs et al., 2014b*). This may be due to the different target host cells of these parasites, with *Plasmodium* infecting terminally differentiated erythrocytes that require extensive remodeling by exported proteins to sustain parasite development, in contrast to *Toxoplasma*, which infects nucleated, dynamic host cells. However, Δ*asp5* parasites displayed a growth defect, demonstrating that

this enzyme is important for the lytic cycle of *Toxoplasma*, in some unknown capacity, at least within HFFs and MEFs.

Whilst ASP5-deficient lines appeared to replicate at a similar rate to wild type tachyzoites, our mouse studies suggest that this enzyme is an important virulence factor. Indeed, we show significant attenuation in mice infected with ~50 live ASP5-deficient tachyzoites, with one mouse surviving beyond 15 days post infection. Furthermore, we show that injection of ~15 live ASP5-deficient tachyzoites is non-lethal to mice and confers protective immunity to lethal challenge. This is in contrast to wild-type RH, which typically show an $LD_{100}$ of 1 parasite. Our data show that ASP5 is important for many cellular and transcriptional changes to the infected host cell and, therefore, this strongly suggests that collectively these changes, even in the highly virulent RH background, are important for *Toxoplasma* virulence in vivo.

The identification of the TEXEL motif and its cleavage by ASP5 provides valuable new insights into the mechanism of host cell modification by *Toxoplasma*. We demonstrate similarities and important differences between this process and the analogous pathway in *Plasmodium* spp. Our work therefore poses new questions as to the functions and mechanisms of protein export between these two parasites as well as other Apicomplexan species of agricultural and medical significance.

## Materials and methods

### Host cell and parasite cultures and transfection

All *Toxoplasma* parasites used in this study are of the 'type I' RH background, either RHΔ*hxgprt* (Δ*hx*), or RHΔ*ku80* (Δ*ku80*). These parasites, and all subsequently derived lines, were cultured in primary HFFs (American Type Culture Collection, ATCC) in Dulbecco's Modified Eagle medium (DME) supplemented with 1% v/v fetal calf serum (FCS) (Invitrogen, Australia) and 1% v/v Glutamax (Invitrogen) (D1). Prior to infection HFFs were grown to confluency in DME supplemented with 10% v/v cosmic calf serum (GE Healthcare, New Zealand) (D10).

Transfection of *Toxoplasma* tachyzoites was performed as previously described (*Soldati and Boothroyd, 1993*). Briefly, parasites were resuspended at $1 \times 10^7$ in 400 µL cytomix and transfected using 15 µg of linear DNA or 50 µg of circular DNA. Linearized DNA was used to tag or modify endogenous loci, while circular DNA was used for transient expression or random integration of ectopic constructs. Electroporation conditions were 1.5 kV, 25 uF and 50 Ω using a Bio-Rad Gene Pulser II (Bio-Rad). Electroporated parasites were transferred to HFFs in D1 immediately after transfection. Parasites expressing the HXGPRT cassette were selected through addition of mycophenolic acid (25 µg/ml) and xanthine (50 µg/ml), the CAT cassette: chloramphenicol (20 µM), the phleomycin cassette: phleomycin (50 µg/ml), the DHFR cassette: 1 µM pyrimethamine and FUDR (5 µM) was used for disruption of *uprt*.

### DNA and plasmids

All primers used in this study are listed in *Supplementary file 1*. Endogenous epitope tagging of ASP5 was achieved through PCR amplification the 3' end of the gene (TGME49_242720), which was cloned into pPR2-HA₃ (*Sheiner et al., 2011*) upstream of a triple HA epitope tag with the -DHFR M2M3 selectable marker to confer pyrimethamine selection. The ectopic expression constructs ASP5$_{WT}$ and ASP5$_{D431A, D682A}$ were synthesized (Epoch Life Science) based on the gene model listed above at toxodb.org (*Gajria et al., 2008*) and cloned into the pHTU-HA₃ vector, which contains the HXGPRT selectable marker, *uprt* disruption fragment and the *Toxoplasma* tubulin promoter (*McCoy et al., 2012*).

The GRA16 allelic swap plasmid was made by Gibson cloning. Flank 1 (F1) was amplified using primers 1 and 2, Flank 2 (F2) amplified using primers 3 and 4. WT GRA16 sequence was synthesized by IDT and amplified using primers 5 and 6. The HXGPRT selectable marker cassette was amplified using primers 7 and 8. pBS plasmid backbone was digested out of pHTG (*McCoy et al., 2012*) using *Bam*HI/*Hind*III. Fragments were combined in equimolar concentrations and reactions undertaken as per manufacturer's instructions. Primers 9 and 10 were used for the mutagenesis of pTKOII-GRA16$_{WT}$-HA to pTKOII-GRA16$_{AAAAE}$-HA as per the manufacturer's protocol (PfuTurbo DNA Polymerase [Agilent Technologies]).

The pTKO-Δ*asp5*-CAT vector was made using primers 11 and 12 to amplify F1 of *asp5*, which was then digested with *Fse*I and *Nsi*I and ligated into the pTKO vector 5' of the HXGPRT selectable cassette. Primers 13 and 14 were used the amplify F2 of *asp5*, digested with *Bgl*II and *Xma*I, then ligated 3' of the HXGPRT selectable marker. The HXGPRT cassette was swapped with the CAT cassette using *Bam*HI/*Hin*dIII. Plasmid was digested with *Nsi*I and *Not*I and co-transfected with pSAG1:: Cas9-U6::sgASP5-1 as described below. PCR of the genomic DNA (gDNA) of the WT *asp5* locus (*Figure 4A*, PCR1) was completed using primers 19 and 20. The PCRs of the resulting Δ*asp5* mutants (PCR2 and PCR3) were performed using primers 21 and 22, and 22 and 23, respectively (the GRA1 promoter drives both GFP and the CAT expression).

Two *asp5*-targeting Cas9 plasmids were generated for this study, one to facilitate integration of double crossover plasmid pTKO-Δ*asp5*-CAT (Δ*ku80*Δ*asp5*, *Figure 4A*) and one for direct disruption (Δ*asp5*$_{CRISPR}$, *Figure 4B*). Briefly, both protospacers were directed towards the first exon and were chosen from toxodb.org if they were specific to the coding region of *asp5* and absent from the rest of the genome (for criteria, see [*Cong et al., 2013*; *Sidik et al., 2014*]). The sequences used for these guides are; gtccgtccccgtctcctcaac and gggtcctgttctgggcagat, respectively. pSAG1::Cas9-U6:: sgASP5-1 was generated in the pSAG1::Cas9-U6::sgUPRT (*Shen et al., 2014*) plasmid using Q5 mutagenesis (Stratagene), using primers 15 and 16 as applied by *Shen et al. (2014)* (pSAG1::Cas9-U6::sgASP5-1). pU6-Universal:sgASP5-2, used to generate the Δ*asp5*$_{CRISPR}$ tachyzoites, was generated in the pU6-Universal plasmid (*Sidik et al., 2014*). This was done by first fusing Cas9 with the 2A skip peptide and mCherry, then subsequent FACS sorting and cloning. The guide was introduced using Q5 mutagenesis using primers 17 and 18.

The GRA24-Myc$_3$-expressing plasmid was generated by codon optimizing *gra24* based on the current gene model (toxodb.org: TGME49_230180) by IDT, then cloned into the pHTU vector described above. Transfection proceeded by linearization within the *uprt* flank and selection for ectopic expression at the *uprt* locus by FUDR selection.

## IFA and antibodies

Parasites were fixed in 4% v/v paraformaldehyde in PBS for 10 min; permeabilized in 0.1% v/v Triton X-100 in PBS and blocked in 3% w/v BSA (Sigma) in PBS for 1 hr. The following antibodies were used in this study: αGAP45 (*Gaskins, 2004*), αSAG1 DG52 (*Burg et al., 1988*), αHA 3F10 (Roche), αc-Myc Y69 (Abcam), αMAF1 (*Pernas et al., 2014*), αCatalase (*Ding et al., 2000*), αGAPDH (Santa Cruz), αMYR1 (In press) and αMyc 9E10 (Sigma). Primary antibodies were diluted in the bovine serum albumin (BSA)/PBS solution for 1 hr, washed, and then incubated with Alexa Fluor-conjugated secondary antibodies (Invitrogen) for 1 hr. 5 μg/ml DAPI was added in the penultimate wash for 5 min and samples were mounted onto microscope slides with Vectashield (Vector Labs). Parasites were imaged using an Allied Precision DeltaVision Elite wide field microscope at 100× magnification (1024 × 1024 pixels) with a CoolSnap2 CCD detector and deconvolved using Softworx V5.0.

## Protease cleavage assays, MS, and enzyme inhibition

Protease cleavage assays were performed using HA-tagged *Toxoplasma* aspartyl protease 5 (ASP5$_{WT}$-HA$_3$) or *P. falciparum* plasmepsin V (PMV-HA) immunopurified from parasite lysates, as described previously (*Boddey et al., 2010*; *Sleebs et al., 2014b*). Briefly, protease bound to agarose was prepared by incubating αHA-agarose (Sapphire Bioscience) in parasite lysates, prepared by sonication in 1% Triton X-100/PBS pH 7.4, for 1 hr before extensive washing in 1% Triton X-100/PBS, followed by storage in PBS. ASP5 cleavage assays comprised of 0.4 μL ASP5$_{WT}$-HA$_3$-agarose in digest buffer (25 mM Tris.HCl, 25 mM MES, pH 5.5; different pH ranges were tested and pH 5.5 was optimal), 0.005% Tween-20, 5 μM TEXEL peptide substrate (GRA16: DABCYL-R-VS**RRLAE**EP-E-EDANS, GRA19: DABCYL-R-VA**RRLSD**RE-E-EDANS, GRA21: DABCYL-R-PV**RELLD**LE-E-EDANS, MYR1: DABCYL-R-DV**RRLSE**QA-EDANS, GRA24: DABCYL-R-ST**RGYHG**GS-E-EDANS, DABCYL-R-AP**RGGLQ**TP-E-EDANS, DABCYL-R-DY**RSLGM**LG-E-EDANS) where residues in bold correspond to the different residues shown in *Figure 3B* (GRA16), *Figure 3C* (GRA19 and 21), *Figure 5D* (MYR1) and *Figure 7C* (GRA24), in 20 μL total volumes. For PMV, digests comprised of 0.2 μL PMV-HA-agarose in digest buffer (25 mM Tris.HCl, 25 mM MES, pH 6.4), 0.005% Tween-20, 5 μM PEXEL peptide substrate (DABCYL-R-NK**RTLAQ**KQ-E-EDANS) where residues in bold correspond to the different

residues shown in *Figure 3A* (KAHRP), in 20 µL total volumes. Samples were incubated at 37°C for 4 hr and processing measured as fluorescence using an Envision plate reader (PerkinElmer) excited at 340 nm and reading emissions at 490 nm. Samples were gently shaken during incubation to disperse protease-agarose. All peptides were synthesized by ChinaPeptides to >85% purity. Products of the incubation of ASP5$_{WT}$-HA$_3$ with DABCYL-R-VS**RRLAE**EP-E-EDANS were detected by a molecular formula algorithm using an Agilent 6200 TOF/6500 series mass spectrometer, as described previously (*Boddey et al., 2010*). Percentage activity of ASP5 and PMV proteases was determined by measuring the maximum fluorescence of cleaved substrate after 4 hr and setting this to 100%, as performed previously (*Boddey et al., 2013*; *Sleebs et al., 2014a*; *2014b*; *Hodder et al., 2015*).

Inhibition of ASP5 by a compound that directly mimics the native GRA16 TEXEL substrate (RRL$_{Statine}$) was performed as described previously (*Sleebs et al., 2014b*). Compounds WEHI-916 (not shown) (*Sleebs et al., 2014b*) and WEHI-586 (synthesis outlined below) were evaluated using the fluorogenic TEXEL cleavage assay described above in a nine-point 1:2 serial dilution of compounds solubilized in dimethyl sulfoxide (DMSO) (1% final concentration). All assay end-points were set within the linear range of activity (approximately 2 hr). IC$_{50}$ values were determined using a nonlinear regression four-parameter fit analysis, where two of the parameters were constrained to 0 and 100%.

## Synthesis of WEHI-586

Analytical thin-layer chromatography was performed on Merck silica gel 60F254 aluminum-backed plates and were visualized by fluorescence quenching under ultraviolet light or by KMnO$_4$ staining. Flash chromatography was performed with silica gel 60 (particle size 0.040–0.063 µm). Nuclear magnetic resonance (NMR) spectra were recorded on a Bruker Avance DRX 300 with the solvents indicated ($^1$H NMR at 300 MHz). Chemical shifts are reported in ppm on the $\delta$ scale and referenced to the appropriate solvent peak. MeOD contains H$_2$O. High-resolution electrospray ionization mass spectroscopies (HRESMS) were acquired by Jason Dang at the Monash Institute of Pharmaceutical Sciences Spectrometry Facility using an Agilent 1290 infinity 6224 TOF LCMS. Column used was RRHT 2.1 x 50 mm 1.8 µm C18. Gradient was applied over the 5 min with the flow rate of 0.5 mL/min. For MS: Gas temperature was 325°C; drying gas 11 L/min; nebulizer 45 psig and the fragmentor 125 V. LCMS were recorded on a Waters ZQ 3100 using a 2996 Diode Array Detector. LCMS conditions used to assess purity of compounds were as follows, column: XBridge TM C18 5 µm 4.6 x 100 mm, injection volume 10 µL, gradient: 10–100% B over 10 min (solvent A: water 0.1% formic acid; solvent B: AcCN 0.1% formic acid), flow rate: 1.5 mL/min, detection: 100–600 nm. All final compounds were analyzed using ultrahigh-performance LC/ultraviolet/evaporative light scattering detection coupled to MS. Unless otherwise noted, all compounds were found to be >95% pure by this method.

The following starting materials were purchased commercially and used without further purification, Cbz-Orn(*N*-Boc)-OH and HCl.NH$_2$-Orn(*N*-Boc)-OMe. HCl,NH$_2$-Sta-NH$_2$(CH$_2$)$_2$ Ph 5 was prepared as previously described. WEHI-916 was prepared as previously described (*Sleebs et al., 2014a*; *2014b*).

## Synthesis of WEHI-586 Step 1: Cbz-Orn(*N*-Boc)-Orn(*N*-Boc)-OMe 1
### General procedure A
Compound numbers refer to the synthesis scheme outlined in *Figure 2—figure supplement 4*. A mixture of Cbz-Orn(*N*-Boc)-OH (500 mg, 1.36 mmol), Et$_3$N (663 µL, 4.76 mmol), NH$_2$Orn(*N*-Boc)-OMe. HCl (463 mg, 1.64 mmol), and HBTU (672 mg, 1.77 mmol), in DMF (5.0 mL) was allowed to stir for 18 hr at 20°C. 10% Citric acid solution was added to the reaction mixture. The solution was extracted with EtOAc (2 x 20 mL). The organic layer was then washed with 10% NaHCO$_3$ solution (20 mL). The organic layer was dried (MgSO$_4$) and the organic layer was concentrated *in vacuo* to obtain an oil. The oil obtained was subjected to silica chromatography gradient eluting with 100% DCM to 10% MeOH/DCM to obtain **1** as a white solid (630 mg, 78%). $^1$H NMR (CDCl$_3$): $\delta$ 7.37 (s, 5H), 7.11 (br s, 1H), 5.60 (br s, 1H), 5.13 (s, 2H), 4.60–4.50 (m, 1H), 4.41–4.32 (m, 1H), 3.74 (s, 3H), 3.35–3.05 (m, 4H), 1.92–1.50 (m, 8H), 1.45 (s, 18H). MS, *m/z* = 595 [M+H]$^+$.

## Synthesis of WEHI-586 Step 2: PhCH₂SO₂-Orn(*N*-Boc)-Orn(*N*-Boc)-OMe 2

A mixture of **1** (0.6 g, 1.01 mmol) and Pd/C (cat.) in MeOH (20 ml) under a hydrogen atmosphere was allowed to stir for 18 hr. The mixture was filtered through Celite and concentrated to dryness *in vacuo*. To the crude oil dissolved in DCM (10 ml), benzylsulfonyl chloride (210 mg, 1.1 mmol and Et₃N (153 µL, 1.1 mmol) was added. The mixture was then allowed to stir for 18 hr at 20°C. The reaction mixture was concentrated to dryness *in vacuo.* The residue obtained was subjected to silica chromatography gradient eluting with 100% DCM to 5% MeOH/DCM to obtain **2** as a white solid (330 mg, 53%). $^1$H NMR (CDCl₃): δ 7.64–7.37 (m, 5H), 7.26 (m, 1H), 5.36 (br s, 1H), 4.56–4.49 (m, 1H), 4.28 (s, 2H), 4.05 (br s, 1H), 3.73 (s, 3H), 3.30–3.00 (m, 4H), 2.00–1.50 (m, 8H), 1.45 (s, 18H). MS, $m/z$ = 615 [M+H]$^+$.

## Synthesis of WEHI-586 Step 3: PhCH₂SO₂-Orn(*N*-Boc)-Orn(*N*-Boc)-OH 3

A mixture of **2** (300 mg, 0.49 mmol), and LiOH hydrate (21 mg, 0.98 mmol) in a mixture of water (3 mL) and THF (5 mL) was allowed to stir for 2 hr at 20°C. 10% citric acid solution was added to the reaction mixture. The solution was extracted with EtOAc (2 × 20 mL). The organic layer was then washed with brine (20 mL). The organic layer was dried (MgSO₄) and the organic layer was concentrated *in vacuo* to obtain **3** as a white solid (220 mg, 75%). $^1$H NMR (CDCl₃): δ 7.45–7.36 (m, 5H), 5.93 (br s, 1H), 5.77–5.60 (m, 1H), 4.90 (br s, 1H), 4.52 (br s, 1H), 4.28 (s, 2H), 4.01–3.90 (m, 2H), 3.20–3.00 (m, 4H), 2.05–1.48 (m, 8H), 1.44 (s, 18H). MS, $m/z$ = 601 [M+H]$^+$.

## Synthesis of WEHI-586 Step 4: PhCH₂SO₂-Orn(*N*-Boc)-Orn(*N*-Boc)-Sta-NH(CH₂)₂Ph 6

General Procedure A was followed using **3** (100 mg, 0.166 mmol), to obtain **6** as a white solid (45 mg, 32%). $^1$H NMR (CDCl₃) (rotamers): δ 7.41–7.15 (m, 10H), 5.10–5.80 (m, 2H), 4.28–4.25 (m, 2H), 4.00–3.80 (m, 3H), 3.53–3.41 (m, 1H), 3.15–2.70 (m, 8H), 2.35–2.20 (m, 2H), 1.80–1.20 (m, 29H), 0.93–0.86 (m, 6H). MS, $m/z$ = 862 [M+H]$^+$.

## Synthesis of WEHI-586 Step 5: PhCH₂SO₂-Arg(*N,N*-diBoc)-Arg(*N,N*-diBoc)-Sta-NH(CH₂)₂Ph 7

A mixture of **6** (40 mg, 0.046 mmol), in 4 N HCl in dioxane (5 mL) was allowed to stir for 30 min at 20°C. The reaction mixture was concentrated to dryness *in vacuo*. The residue was dissolved in DCM (10 ml) and Et₃N (38 µL, 0.276 mmol) was added. The solution was stirred vigorously for 5 min. *N, N′*-bis-Boc-1-guanylpyrazole (31 mg, 0.101 mmol) was added and the solution was left to stir for 12 hr. 10% citric acid solution was added to the reaction mixture. The solution was extracted with DCM (2 × 15 mL). The organic layer was then washed with 10% NaHCO₃ solution (20 mL). The organic layer was dried (MgSO₄) and the organic layer was concentrated *in vacuo* to obtain an oil. The oil was subjected to silica chromatography gradient eluting with 100% DCM to 10% MeOH/ DCM to obtain **7** as a white solid (35 mg, 66%). $^1$H NMR (CDCl₃) (rotamers): δ 8.60–8.30 (m, 3H), 7.38–7.21 (m, 11H), 6.68 (br s, 1.5H), 6.40 (br s, 1H), 4.40–4.26 (m, 4H), 4.00–3.30 (m, 10H), 2.85–2.75 (m, 2H), 1.85–1.10 (m, 47H), 0.89–0.87 (m, 6H). MS, $m/z$ = 1146 [M+H]$^+$.

## Synthesis of WEHI-586 Step 6: PhCH₂SO₂-Arg-Arg-Sta-NH(CH₂)₂Ph. 2TFA - WEHI-586

A mixture of **7** (35 mg, 0.03 mmol) in TFA (0.5 mL) and DCM (1 mL) was allowed to sit for 18 hr at 20°C. The reaction mixture was concentrated to dryness *in vacuo*. The oil was triturated with Et₂O and filtered off, washing with Et₂O, to obtain to obtain **WEHI-586** as a white solid (26 mg, 87%). $^1$H NMR (MeOD): δ 7.47–7.17 (m, 10H), 4.40–4.29 (m, 4H), 3.98–3.77 (m, 2H), 3.45–3.19 (m, 8H), 2.83–2.73 (m, 2H), 1.74–1.61 (m, 11H), 1.00–0.85 (m, 6H). HRESMS found: (M+H) 745.4197; C₃₅H₅₆N₁₀O₆S requires (M+H), 745.4183.

## ASP5 modeling

Homology models for the complex of *Toxoplasma* ASP5 with GRA16 (residues S-R3R2L1A1′E2′-E) were generated using the MODELLER program (version 9.14) (*Eswar et al., 2006*) using structures of *P. vivax* PMV in complex with WEHI-842 and *P. vivax* plasmepsin IV in complex with Pepstatin A

as templates (PDB codes 4ZL4 [*Hodder et al., 2015*] and 1QS8 [*Bernstein et al., 2003*], respectively). Restraints were included to ensure the carbonyl oxygen of the L3 residue of the substrate was within hydrogen bonding distance of the PvPMV catalytic aspartic acid (D531). Mutations were introduced using the YASARA program (http://www.yasara.org); hydrogen atoms were added to complete atomic valencies and the geometries minimized using the AMBER force field (*Duan et al., 2003*). An evaluation of the binding free energy was carried out using the AutoDock potential (*Morris et al., 1998*). In its original formulation, the AutoDock potential includes terms that represent the entropic penalty for restriction of conformational freedom and desolvation of the ligand only. Here, we have included these two components for both ligand (wild-type GRA16 and mutations) and receptor (ASP5) and consequently reduced the contribution to the total free energy of interaction of each by half. AMBER all-atom partial atomic charges were used to calculate the electrostatic interaction energy. Ionizable residues were assumed to be in their standard state at neutral pH except for the catalytic aspartic acid, which was neutral.

## Transmission electron microscopy

*Toxoplasma* tachyzoites were prepared for TEM analysis as described (*Breinich et al., 2009*). Briefly, HFFs were infected at a multiplicity of infection (MOI) of 5:1 for 16 hr, washed twice with PBS, dislodged with trypsin- ethylenediaminetetraacetic acid (Gibco), quenched with cold PBS and pelleted at 1200 *g* for 5 min. PBS-trypsin was replaced with 2.5% glutaraldehyde (Electron Microscopy Sciences) in 0.1 M sodium phosphate buffer. Samples were fixed in osmium tetroxide, dehydrated in ethanol, treated with propylene oxide and embedded in epoxy resin. Sections were stained with uranyl acetate and lead citrate and examined on a Jeol 1200 EX electron microscope.

## Generation of MEFs expressing MTS-GFP

SV40-immortalized MEFs derived from C57BL/6 E14.5 embryos were retrovirally-infected with the MTS of Smac/DIABLO (*Verhagen et al., 2000*) fused to the C-terminus of GFP in an internal ribosome entry site-hygromycin expression vector.

## Transcriptional analysis of host cells

HFFs were passaged and grown in D10 media until they reached confluency. Following this, HFFs were transferred into D1 media and left as uninfected (no parasites) or infected at a MOI of 5 with either RHΔ*ku80* (WT) or Δ*ku80*Δ*asp5* (Δ*asp5*) tachyzoites for 18 hr. HFFs were washed with PBS to remove uninvaded parasites and dislodged with trypsin. Total host and parasite RNA was extracted using the RNeasy kit (Qiagen). Three independent biological replicates of each condition were obtained. DNA libraries were prepared using the Illumina TruSeq v2 protocol and sequenced on an Illumina HiSeq 2000 at the Australian Genome Research Facility (AGRF), Melbourne. On average, 25.5 million 100 bp single-end reads were obtained for each sample. The reads were aligned to the human genome (hg19) using the Rsubread aligner (*Liao et al., 2013*). The number of fragments overlapping each Entrez gene were counted using featureCounts (*Liao et al., 2014*) and NCBI RefSeq annotation, build 38.1. Differential expression analyses were performed using the Bioconductor packages edgeR (*Robinson et al., 2010*) and limma (*Ritchie et al., 2015*). All genes that did not achieve a count per million of 0.4 in at least 3 samples were deemed to be unexpressed and subsequently filtered from the analysis. Additionally, genes with no official symbol in the NCBI gene information file were removed. Following filtering, 15,018 genes remained for the downstream analysis. Compositional differences between samples were normalized using the trimmed mean of M-values method (*Robinson et al., 2010*). All counts were then transformed to log2-counts per million (logCPM) with associated precision weights using voom (*Law et al., 2014*). Differential expression for the three comparisons, WT infected versus uninfected, Δ*asp5* infected versus uninfected, and Δ*asp5* infected versus WT infected, was assessed using empirical Bayes moderated t-statistics (*Smyth, 2004*). Genes were considered to be differentially expressed if they attained a false discovery rate of 0.05. Gene ontology analysis used the goana function. Shrunk log2-fold-changes for plotting were computed using edgeR's predFC function with prior count set to 3. The barcode plot was drawn with limma's barcodeplot function and the correlation of the Δ*gra16* gene sets with the RNA-seq data was evaluated using a directional roast gene set test with 10,000 rotations (*Wu et al.,*

*2010*). This data has been deposited in NCBI's Gene Expression Omnibus (GEO) under accession number GSE73986.

## Virulence studies

All animal experiments were performed in accordance with regulations outlined by The Walter and Eliza Hall Institute's Animal Ethics Committee. Wild-type (RH$\Delta$*hxgprt*), $\Delta asp5_{CRISPR}$ and $\Delta asp5_{CRISPR}$: ASP5$_{WT}$-HA$_3$ parasites were grown in HFFs, harvested, and counted. Doses were either 100 tachyzoites in 200 µL of PBS (determined by plaque assay to be 15 ± 3 tachyzoites) or 200 parasites in 200 µL PBS (~50 live parasites). All tachyzoites were intraperitoneally injected into 6 × 6–8-week-old C57BL/6 mice. Mice were monitored daily and sacrificed when determined moribund. Sero-conversion was monitored by using serum collected from mice at 14 days post infection and used in Western blot (1:500 dilution) against purified tachyzoites and uninfected HFF lysates.

## Acknowledgements

We would like to thank the expert help of Liana Mackiewicz, Carolina Alvarado and Lisa Reid from WEHI Bioservices and Stephen Wilcox from the Ian Potter Centre for Genomics and Personalised Medicine at WEHI. We would also like to thank Kelly Rogers, Lachlan Whitehead and Mark Scott from WEHI's Centre for Dynamic Imaging, and Alan Cowman for his support and useful discussions.

## Additional information

### Funding

| Funder | Author |
| --- | --- |
| National Health and Medical Research Council | Justin A Boddey Christopher J Tonkin |
| Australian Research Council | Justin A Boddey Christopher J Tonkin |
| National Institutes of Health | John C Boothroyd |

The funders had no role in study design, data collection and interpretation, or the decision to submit the work for publication.

### Author contributions

MJC, BES, ADU, MWP, DJPF, ME, SL, RJS, GD, GKS, BJS, SLM, JCB, JAB, CJT, Conception and design, Acquisition of data, Analysis and interpretation of data, Drafting or revising the article; AG, Acquisition of data, Analysis and interpretation of data, Drafting or revising the article, Contributed unpublished essential data or reagents; MF, NDM, MTO, Conception and design, Acquisition of data, Analysis and interpretation of data, Drafting or revising the article, Contributed unpublished essential data or reagents

### Author ORCIDs

Michael J Coffey, http://orcid.org/0000-0002-8025-4558
Brad E Sleebs, http://orcid.org/0000-0001-9117-1048
David JP Ferguson, http://orcid.org/0000-0001-5045-819X
Gordon K Smyth, http://orcid.org/0000-0001-9221-2892
Justin A Boddey, http://orcid.org/0000-0001-7322-2040
Christopher J Tonkin, http://orcid.org/0000-0002-7036-6222

### Ethics

Animal experimentation: All animal experiments complied with the regulatory standards of and were approved by the Walter and Eliza Hall Institute Animal Ethics Committees under approval number 2014.019.

## Additional files

### Supplementary files
• Supplementary file 1. Primers used in this study.

### Major datasets
The following datasets were generated:

| Author(s) | Year | Dataset title | Dataset URL | Database, license, and accessibility information |
|---|---|---|---|---|
| Coffey MJ, Tonkin CJ, Garnham AL | 2015 | An aspartyl protease defines a novel pathway for export of Toxoplasma proteins into the host cell | http://www.ncbi.nlm.nih.gov/geo/query/acc.cgi?acc=GSE73986 | Publicly available at the NCBI Gene Expression Omnibus (Accession no: GSE73986). |

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
