## [Decision Letter]

Thank you for submitting your work entitled "An aspartyl protease defines a novel pathway for export of *Toxoplasma* proteins into the host cell" for consideration by *eLife*. Your article has been reviewed by two peer reviewers, and the evaluation has been overseen by a Reviewing Editor and Detlef Weigel as the Senior Editor.

The reviewers have discussed the reviews with one another and the Reviewing editor has drafted this decision to help you prepare a revised submission.

Summary:

This paper reports the identification in *Toxoplasma* of the resident Golgi protease ASP5 that appears to be necessary for trafficking and secretion of a large number of dense granule proteins of diverse function. The work spans the spectrum from molecular characterization of the requirements for ASP5 activity to studying its impact on well-characterized effectors that manipulate the host cell. Overall, your study is well performed, timely, and important. Although the Bougdour group just published related work earlier this month, your manuscript contains additional important data that significantly deepen our understanding of ASP5. However, there are several critical points that need to be addressed during revision.

Essential revisions:

1) It is essential that murine virulence studies are performed to demonstrate unequivocally that dense granule secretion is important to virulence, since it is not essential for in vitro growth. If ASP5 is required for the secretion of multiple effectors and/or a large number of changes in gene expression in the host, one might expect a dramatic difference in vivo (and the plaque growth phenotype, if confirmed in multiple ASP5 mutants and reverted after complementation, would also predict a significant deficiency in vivo). While RH is not the best system in which to study this, slight alterations in pathogenesis can be detected in the form of reduced time to death and/or reduced morbidity (weight loss, parasite burden). Given that the strains are in hand, it should not be difficult to carry out these experiments.

2) Many of the differences seen with the RNA-seq data could simply be due to differences in parasite growth rate. If this cannot be resolved, Figure 8 should be removed. Also Figure 8 is very difficult to follow as it is described. Could it be displayed in a simpler fashion or clarified? From the start the labels are confusing, how it is displayed and explained needs to be improved.

3) Subheading “ASP5 modeling reveals key interactions with the TEXEL of GRA16”: The ligand docking studies are interesting but are still based on a threading model rather than a solved structure for ASP5. Therefore it is important that more control experiments are performed using a variety of peptides, showing that RRLAE and RTLAE bind differently. This is interesting, but what about RRL, that is, what is the effect of mutating only residues predicted to be unimportant?

4) Figure 4 and text: Please demonstrate that rescue of the growth phenotype can be rescued by complementation. This seems to be important to ensure that the phenotype is really associated with *asp5* and not off-target effects of other plasmid insertions or CRISPR. Does the second mutant (discussed later in the manuscript) also make smaller plaques? Additionally, what is the impact of ASP5 deletion during infections in mice?

5) Figure 5 and text associated with it: While the IFA suggests that there is residual c-myc nuclear translocation in ASP5 mutant parasites, we would like to see a protein blot from uninfected cells as well. Without it one cannot say that it was "induced" as is mentioned in the text. Also the word "drastically" in the figure legend is not quantitative. Please also discuss the doublet band in wild type and putative cross-reactive protein in the MYR 1 blots in the text in addition to the figure legend.

6) Subheading “ASP5 influences host mitochondrial recruitment to the parasitophorous vacuole membrane”: It seems you quantified the numbers of dense granules or assessed Golgi ultrastructure in a quantitative way; this should be reported. If not, then this statement should be qualified or removed.

---

## [Author Response]

*1) It is essential that murine virulence studies are performed to demonstrate unequivocally that dense granule secretion is important to virulence, since it is not essential for in vitro growth. If ASP5 is required for the secretion of multiple effectors and/or a large number of changes in gene expression in the host, one might expect a dramatic difference in vivo (and the plaque growth phenotype, if confirmed in multiple ASP5 mutants and reverted after complementation, would also predict a significant deficiency in vivo). While RH is not the best system in which to study this, slight alterations in pathogenesis can be detected in the form of reduced time to death and/or reduced morbidity (weight loss, parasite burden). Given that the strains are in hand, it should not be difficult to carry out these experiments.*

We have now performed them and they can be found in Figure 10. We injected 100 wildtype, *△asp5_CRISPR_* and *△asp5_CRISPR_*:ASP5_WT_-HA_3_ into groups of four C57BL/6 mice. The viable effective dose was found to be 15 ± 3 tachyzoites by plaque assay. Strikingly, all mice infected with wildtype and *△asp5_CRISPR_*:ASP5_WT_-HA_3_ (complemented line) succumbed to infection by day 8 whilst mice infected with *△asp5_CRISPR_* all survived to the completion (day 20) of the experiment. Surviving animals were monitored for sero-conversion to *T. gondii* by Western blot (Figure 10—figure supplement 1). This is a profound attenuation in virulence for the hypervirulent RH strain.

*2) Many of the differences seen with the RNA-seq data could simply be due to differences in parasite growth rate. If this cannot be resolved, Figure 8 should be removed. Also Figure 8 is very difficult to follow as it is described. Could it be displayed in a simpler fashion or clarified? From the start the labels are confusing, how it is displayed and explained needs to be improved.*

The replication rates of wildtype, *△asp5_CRISPR_* and *△asp5_CRISPR_*:ASP5-HA parasites have now been compared in experimental replicates using standard assays and are shown in Figure 4. In doing this we could see only very small changes in replication rates between lines. This shows that replication rate is highly unlikely to be the cause of the profound changes in the transcriptional pattern of host cells induced by *Toxoplasma* lines. As such, we have opted to retain the RNA-seq data in the manuscript.

3) Subheading “ASP5 modeling reveals key interactions with the TEXEL of GRA16”: The ligand docking studies are interesting but are still based on a threading model rather than a solved structure for ASP5. Therefore it is important that more control experiments are performed using a variety of peptides, showing that RRLAE and RTLAE bind differently. This is interesting, but what about RRL, that is, what is the effect of mutating only residues predicted to be unimportant?

More controls have now been undertaken in this analysis. In addition to mutating the RR residues in the TEXEL, of GRA16, we have now explored the consequence of mutating the remaining residues, LAE, on binding to ASP5 in the dynamic model. Mutation of leucine to valine (L>V) reduces the calculated binding energy by 6 kJ/mol, in line with this residue playing an important role in processing. However mutation of the fourth TEXEL residue, alanine to valine (A>V) had almost no effect on binding energy in agreement with a lack of sensitivity at this position. Finally, mutation of glutamine at position five of the TEXEL to alanine (E>A) had caused an 11 kJ/mol reduction in binding affinity, in contrast to the increase in ASP5 activity observed in vitro (Figure 3). It is possible that the glutamine reside at position 6 (i.e. RRLAEE) can act as a surrogate for the loss of glutamine at position 5 in this interaction. These data have now been discussed in the text (paragraph three, subheading “ASP5 modeling reveals key interactions with the TEXEL of GRA16”) and are included in Figure 3. The text has also been updated to reflect these additional experiments.

*4) Figure 4 and text: Please demonstrate that rescue of the growth phenotype can be rescued by complementation. This seems to be important to ensure that the phenotype is really associated with* asp5 *and not off-target effects of other plasmid insertions or CRISPR. Does the second mutant (discussed later in the manuscript) also make smaller plaques? Additionally, what is the impact of ASP5 deletion during infections in mice?*

This has now been performed and added as Figure 4. *△asp5_CRISPR_*, like *△ku80△asp5*, also makes smaller plaques which is rescued upon complementation with ASP5. In Figure 4 we also show that the replication rate of the *△asp5_CRISPR_* parasites is not the cause of the reduced plaque size.

*5) Figure 5 and text associated with it: While the IFA suggests that there is residual c-myc nuclear translocation in ASP5 mutant parasites, we would like to see a protein blot from uninfected cells as well. Without it one cannot say that it was "induced" as is mentioned in the text. Also the word "drastically" in the figure legend is not quantitative (and words like this are always open to interpretation). Please also discuss the doublet band in wild type and putative cross-reactive protein in the MYR 1 blots in the text in addition to the figure legend.*

We have repeated the c-myc IFAs and quantitated their signal using confocal imaging. (Figure 5). You can now clearly see that uninfected cells do not express c-myc, whereas cells infected with wild type parasites do. This reflects the previous work by Franco et al., 2014, Eukaryotic Cell, which shows that *Toxoplasma* upregulates c-myc in the host cell. We furthermore show that we see c-myc induction is lost upon deletion of ASP5. It is also important to mention that the induction of c-Myc by *Toxoplasma* tachyzoites has been extensively studied previously and reported in Franco et al., 2014, Eukaryotic Cell, which we cite throughout the appropriate section.

We have also removed the word ‘drastically’ and have now referred to the doublet within the text.

*6) Subheading “ASP5 influences host mitochondrial recruitment to the parasitophorous vacuole membrane”: It seems you quantified the numbers of dense granules or assessed Golgi ultrastructure in a quantitative way; this should be reported. If not, then this statement should be qualified or removed.*

We thank the reviewers for highlighting this confusing wording. This statement has now been removed from the manuscript.